# MedMMV: A Controllable Multimodal Multi-Agent Framework for Reliable and Verifiable Clinical Reasoning

## Abstract

Recent progress in multimodal large language models (MLLMs) has demonstrated promising performance on medical benchmarks and in preliminary trials as clinical assistants. Yet, our pilot audit of diagnostic cases uncovers a critical failure mode: instability in early evidence interpretation precedes hallucination, creating branching reasoning trajectories that cascade into globally inconsistent conclusions. This highlights the need for clinical reasoning agents that constrain stochasticity and hallucination while producing auditable decision flows. We introduce MedMMV, a controllable multimodal multi-agent framework for reliable and verifiable clinical reasoning. MedMMV stabilizes reasoning through diversified short rollouts, grounds intermediate steps in a structured evidence graph under the supervision of a Hallucination Detector, and aggregates candidate paths with a Combined Uncertainty scorer. On six medical benchmarks, MedMMV improves accuracy by up to 12.7% and, more critically, demonstrates superior reliability. Blind physician evaluations confirm that MedMMV substantially increases reasoning truthfulness without sacrificing informational content. By controlling instability through a verifiable, multi-agent process, our framework provides a robust path toward deploying trustworthy AI systems in high-stakes domains like clinical decision support.

## 1 Introduction

Recent frontier multimodal large language models (MLLMs), such as Claude-Sonnet-4 (Anthropic, 2025) and GPT-5 (OpenAI, 2025), are beginning to translate strong general reasoning abilities into healthcare applications. When combined with techniques like chain-of-thought (CoT) prompting (Wei et al., 2023), these systems have achieved state-of-the-art performance on medical question answering benchmarks, narrowing the gap to human expert performance (Singhal et al., 2025; Bhayana et al., 2023; Sandmann et al., 2025; Bedi et al., 2025). Beyond static benchmarks, MLLMs are increasingly being evaluated as agentic clinical assistants in realistic settings, including randomized, double-blind standardized-patient trials, with considerable utility gains (Tu et al., 2024; 2025; Schmidgall et al., 2025; Qiu et al., 2024; Zhu et al., 2025b).

Despite this progress, deploying agent-based systems in real-world healthcare is challenging due to the need for extreme reliability (Zhu et al., 2025a). As illustrated in Figure 1, existing systems face two primary issues that are largely overlooked by benchmarks focused on final-answer accuracy: (1) *instability*, where decisions are highly sensitive to noisy or context-dependent data, and minor perturbations can lead to different outcomes (Singhal et al., 2023a); and (2) *hallucinated reasoning chains*, where models generate fabricated or unsupported facts to justify their conclusions.

To investigate this, we conducted a pilot audit of 100 clinical diagnostic cases, generating multiple responses per case and applying two probes: the random guess measure (RGM) for decision instability and the cross-modal hallucination rate (CMHR) to quantify evidence fabrication. Our audit revealed a critical failure pattern: instability often precedes hallucination. Specifically, when a model's reasoning was unstable, indicated by high prior entropy across rollouts, it frequently switched its majority-voted reasoning path. Such a switch increased the immediate likelihood of hallucination by over 13%, ultimately leading to globally inconsistent conclusions.

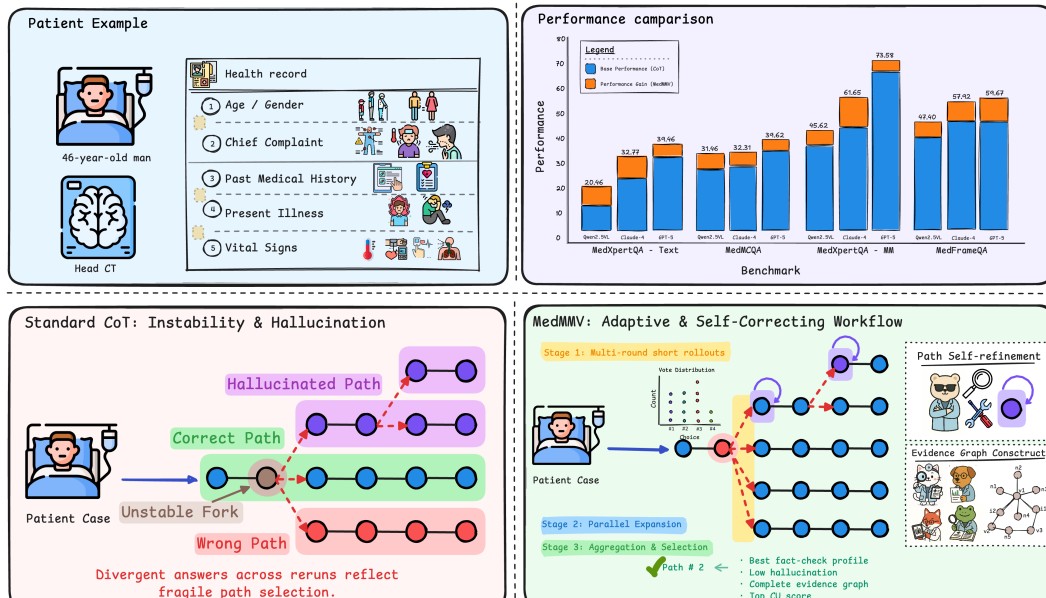

Figure 1: An illustration of MedMMV compared to standard CoT reasoning. *Top-left*: An example patient case as input. *Top-right*: Performance comparison between MedMMV and CoT. *Bottom-left*: Standard CoT exhibits instability, frequently diverging at early reasoning forks and following incorrect or hallucinated paths, leading to unreliable diagnostic conclusions. *Bottom-right*: MedMMV mitigates these issues through a three-stage workflow centered on parallel exploration and evidence-graph-grounded verification, ensuring verifiable and clinically reliable reasoning.

This empirical evidence motivates our central research question: *How can we design clinical agent systems that explicitly constrain stochasticity and hallucination while exposing auditable representations of their decision-making process?* We present MedMMV, a multimodal multi-agent framework that directly addresses the error modes observed in our audit. Rather than committing to a single reasoning trajectory early, MedMMV explores diverse diagnostic hypotheses through multi-round short rollouts, reducing path instability at uncertain decision points. Each path is then refined under the supervision of a Hallucination Detector, which grounds reasoning steps in an evidence graph and prevents local errors from cascading. The refined candidates are finally aggregated by a Combined Uncertainty (CU) Scorer, yielding diagnoses that are not only accurate but also verifiably supported.

To validate the effectiveness of MedMMV, we conduct extensive experiments on six public medical benchmarks spanning both multimodal VQA and text-based QA. As shown in Figure 1, Med-MMV consistently improves accuracy, achieving gains of 7.9% on MedXpert-MM and 12.7% on MedFrameQA when using GPT-5 as the executor. Reliability-oriented metrics further highlight that MedMMV improves truthfulness (TRUE) while maintaining informativeness (INFO), yielding TRUE×INFO gains of 8–12% across datasets. Beyond automated evaluation, physician studies confirm MedMMV's clinical reliability, with our framework achieving a TRUE score of 4.36 versus 3.49 for CoT. Ablation studies demonstrate that hallucination control and uncertainty-aware aggregation are the main contributors; removing the CU scorer drops accuracy by 11% and TRUE×INFO by 13%.

In summary, our contributions are threefold: (1) **Empirical Analysis**: We identify a critical failure mechanism where reasoning instability serves as a direct precursor to hallucination, showing how early-stage stochasticity in evidence interpretation cascades into global inconsistency. (2) **MedMMV Framework**: We propose a controllable, multimodal multi-agent reasoning system that integrates diversified rollouts, evidence-grounded refinement, and uncertainty-aware aggregation to mitigate these errors. (3) **Comprehensive Evaluation**: Through experiments on six benchmarks, physician studies, and ablations, we demonstrate state-of-the-art reliability and provide new insights into building trustworthy clinical reasoning systems.

## 2 RELATED WORK

**Consistency and hallucination in multimodal clinical reasoning.** Ensuring process fidelity is especially challenging in high-stakes, multimodal clinical settings, where sparse or misaligned evidence can trigger random guessing and cross-modal hallucinations (Liu et al., 2024; Bai et al., 2025b). Existing medical MLLMs like LLaVA-Med (Li et al., 2023a) and Med-Flamingo (Moor et al., 2023) show progress on image-text QA but still lack robust uncertainty calibration and step-level verifiability. While self-consistency reduces variance, it cannot guarantee evidence-faithful reasoning (Wang et al., 2023). Benchmarks such as POPE (Li et al., 2023c), HallusionBench (Guan et al., 2024), Med-HALT (Pal et al., 2023), and ConBench (Zhang et al., 2024) expose hallucination patterns and assess trajectory-level consistency. Distinct from these probes, our analysis highlights how early stochasticity in evidence interpretation branches reasoning trajectories and cascades into global inconsistency, echoing the "snowballing" effect in multimodal hallucinations (Zhong et al., 2024). To capture this process, we employ dispersion across paths (RGM) and cross-modal hallucination rate (CMHR) not as end goals but as early-warning signals of this cascading dynamic.

**Controllable reasoning and process-level regulation.** A growing body of literature shows that process-level supervision and verification, rather than outcome-only scoring, improves reliability by checking intermediate steps and rationales (Lightman et al., 2023; Stiennon et al., 2022). In parallel, uncertainty estimation enables models to "know when they know" (Kadavath et al., 2022), providing a principled basis for gating and adjudication. Exploration strategies such as CoT (Wei et al., 2023), self-consistency (Wang et al., 2023), Tree-of-Thoughts (Yao et al., 2023), and multi-agent debate expand reasoning diversity (Kim et al., 2024; Li et al., 2023b; Chen et al., 2024). However, these methods remain prone to correlated errors and weak cross-modal alignment. MedMMV operationalizes a controllable variant of the multi-agent paradigm with three distinct levers: (1) uncertainty-aware hypothesis generation that seeds diversified yet calibrated paths; (2) independent, evidence-grounded verification for each path via process supervision; and (3) quantitative aggregation via a combined uncertainty scorer that selects the most robustly supported conclusion.

As summarized in Table 1, MedMMV distinguishes itself by integrating parallel exploration with active supervision and evidence grounding, producing fully traceable reasoning paths and enabling a verifiable approach to clinical decision-making.

Table 1: Comparison of MedMMV with representative prior methods.

| System | Data Modality | | Core Reasoning Engine | | | Structured Reasoning & Validation | | | |
| --- | --- | --- | --- | --- | --- | --- | --- | --- | --- |
| | Text | Image | Self-Revision | Evidence Grounding | Halluc. Control | Parallel Exploration | Active Supervision | Consensus & Aggregation | Traceable Reasoning Path |
| *Reasoning Methods* | | | | | | | | | |
| CoT (Wei et al., 2023) | ✓ | ✗ | ✗ | ✗ | ✗ | ✗ | ✗ | ✗ | ✗ |
| MedPaLM (Singhal et al., 2023b) | ✓ | ✓* | ✓ | ✓ | ✓ | ✗ | ✗ | ✓** | ✗ |
| *Multi-Agent Systems* | | | | | | | | | |
| ClinicalAgent (Yue et al., 2024) | ✓ | ✗ | ✓ | ✗ | ✗ | ✗ | ✗ | ✗ | ✗ |
| MDAgents (Kim et al., 2024) | ✓ | ✓ | ✓ | ✗ | ✗ | ✗ | (✓) | (✓) | ✗ |
| ColaCare (Wang et al., 2025b) | ✓ | ✗ | ✓ | ✓ | ✗ | ✗ | (✓) | ✓ | ✗ |
| ReConcile (Chen et al., 2024) | ✓ | ✗ | ✓ | ✗ | ✗ | ✗ | (✓) | ✓ | ✗ |
| *Agentic Workflow Automation* | | | | | | | | | |
| AFlow (Zhang et al., 2025) | ✓ | ✗ | ✓ | ✗ | ✗ | ✗ | ✗ | ✗ | ✗ |
| ADAS (Hu et al., 2025) | ✓ | ✗ | ✓ | ✗ | ✗ | ✗ | ✗ | ✗ | ✗ |
| **MedMMV (ours)** | ✓ | ✓ | ✓ | ✓ | ✓ | ✓ | ✓ | ✓ | ✓ |

*MedPaLM-M, a variant, handles multiple modalities. **Achieved via self-consistency. (✓) Denotes partial or implicit support.

## 3 PRELIMINARY STUDY: FROM INSTABILITY TO HALLUCINATION

We present two empirical diagnostics, the random guess measure (RGM) and cross-modal hallucination rate (CMHR), to expose reasoning instability. We conduct a pilot audit on medical QA and VQA (each $n \approx 50$) and connect the findings to prior theory (Kalai et al., 2025). The evidence establishes two insights that motivate our framework: (1) randomness and hallucination co-occur within a single reasoning trajectory, and (2) local evidence errors propagate into globally inconsistent conclusions.

### 3.1 QUANTITATIVE MEASURES

To characterize how an LLM behaves under repeated sampling, we introduce two complementary diagnostics. Both metrics are computed across multiple independent generations of the same question,

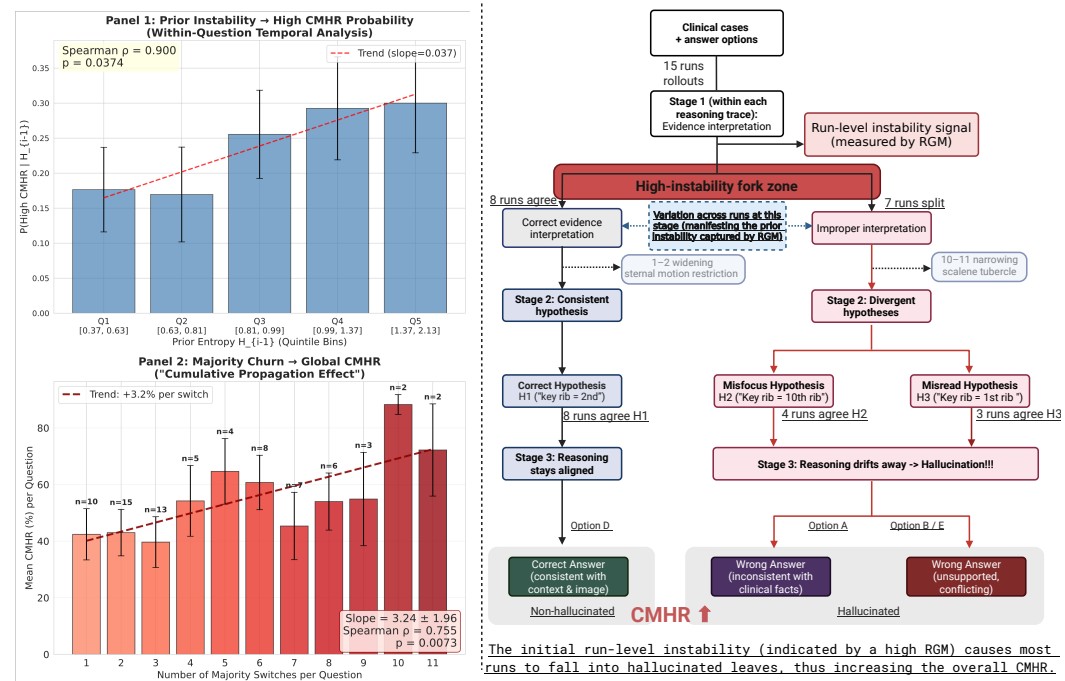

Figure 2: Instability breeds hallucination. *Left (Panels 1–2):* Prior instability strongly predicts hallucination risk. (1) Higher entropy at step $i - 1$ increases the probability of high CMHR at step $i$. (2) Majority-path churn accumulates globally and correlates with higher mean CMHR. *Right:* A multi-rollout case study illustrates how rollout-level prior instability (measured by RGM) manifests within a single generation. At the evidence interpretation stage, this prior cross-rollout instability makes it more likely to split into a high-instability fork (red). The unstable interpretations propagate into divergent hypotheses and eventually lead to hallucinated answers (red leaves), whereas the stable cluster of rollouts (blue/green) maintains a consistent hypothesis and arrives at the correct conclusion.

and are used to quantify (i) the stability of the model's decisions across rollouts and (ii) the reliability of the produced answers. The accompanying case study (right side of Fig. 2) provides an intuitive visualization of the cross-rollout phenomena captured by these measures.

**Random guess measure (RGM).** For a question $q$ with an option set $\mathcal{O}_q$ of size $n_q$, we sample $k_q$ independent generations with categorical choices $c_{q,i} \in \mathcal{O}_q$, for $i = 1, \ldots, k_q$. Before round $i \geq 2$, we define the empirical prior over options as:

$$p_{q,i-1}(o) = \frac{1}{i-1} \sum_{t=1}^{i-1} \mathbf{1}[c_{q,t} = o], \quad o \in \mathcal{O}_q \tag{1}$$

The entropy of this distribution, where $H_2(\mathbf{p}) = \log_2 n_q - D_{\mathrm{KL}}^{(2)}(\mathbf{p} \parallel \mathbf{u}_{n_q})$, quantifies how dispersed earlier generations are. High entropy indicates pronounced rollout-to-rollout disagreement. In Fig. 2, this cross-rollout dispersion is visualized as a wide fork in the evidence-interpretation stage, where different rollouts commit to incompatible interpretations of the same clinical cue. The question-level instability is defined as:

$$\mathrm{RGM}_q = \frac{1}{k_q - 1} \sum_{i=2}^{k_q} H_2(\mathbf{p}_{q,i-1}), \qquad \mathrm{RGM}_q^{\mathrm{early}} = \frac{1}{L} \sum_{i=2}^{L+1} H_2(\mathbf{p}_{q,i-1}). \tag{2}$$

Higher RGM values indicate divergent decisions across rollouts, signifying unstable path selection akin to stochastic guessing. This instability often precedes and helps explain the propagation of errors observed in hallucinated paths.

**Cross-modal hallucination rate (CMHR).** Each generation is rated by a panel of models $\mathcal{M} = \{\mathrm{GPT\text{-}4o}, \mathrm{Claude\text{-}Sonnet\text{-}4}, \mathrm{Gemini}\}$ on truthfulness $T_{i,m} \in [1, 5]$, informativeness $I_{i,m} \in [1, 5]$,

and text-vision consistency $C_{i,m} \in [0, 1]$ if an image is present. Averaging across these raters yields:

$$\bar{T}_i = \tfrac{1}{|\mathcal{M}|} \sum_m T_{i,m}, \quad \bar{I}_i = \tfrac{1}{|\mathcal{M}|} \sum_m I_{i,m}, \quad \bar{C}_i = \tfrac{1}{|\mathcal{M}|} \sum_m C_{i,m}. \tag{3}$$

Following Lin et al. (2022) and Wang et al. (2025a), we combine three raters into a reliability score, and define the cross-modal hallucination rate:

$$\text{Rel}_i = \left( \tfrac{\bar{T}_i}{5} \right) \left( \tfrac{\bar{I}_i}{5} \right) \bar{C}_i, \qquad \text{CMHR} = 100 \cdot \left( 1 - \mathbb{E}_i[\text{Rel}_i] \right). \tag{4}$$

Lower reliability corresponds to answers that contradict visual evidence, misinterpret clinical facts, or lack informational adequacy. In the case study figure, these appear as red hallucinated leaves, final outputs arising from misfocused or misread evidence. A higher CMHR reflects a greater fraction of such low-credibility generations.

## 3.2 PILOT AUDIT

We perform repeated CoT sampling on 100 medical items (50 QA, 50 VQA), all with RGM $> 0$, totaling 1,500 generations (15 rounds per question). A majority-path switch occurs when the round-$i$ majority-voted reasoning branch differs from round $i-1$; the per-question churn is the count of such switches. Analysis details appear in Appendix A. Figure 2 visualizes all results.

**Finding A: Randomness and hallucination co-occur in trajectories.** Conditioning on the prior entropy $H_{i-1}$, the probability of high CMHR at the next round increases monotonically across entropy quintiles (slope $\approx +0.037$ per bin; Spearman $\rho \approx 0.90$, $p \approx 0.037$; Figure 2, Panel 1). Mechanistically, a majority-path switch produces an immediate risk jump, where the chance of high CMHR rises from $\sim 0.21$ (no switch) to $\sim 0.34$ (switch), a 13.3% difference (Fisher $p \approx 0.000$; Figure 2, Panel 3). The case study on the right corroborates these micro-dynamics. A high-RGM fork (Step 2) spawns divergent hypotheses, several terminating in hallucinated leaves due to mis-mapped rib features, while the stable path remains consistent.

**Finding B: Local evidence errors propagate to global inconsistency.** Aggregating by question, churn predicts higher CMHR, where mean CMHR increases with the number of switches (Pearson $r = 0.270$, $p = 0.0193$; Figure 2, Panel 2). Panel 4 shows temporal structure where high-churn items accumulate entropy over rounds, while low-churn items remain comparatively stable. In the case tree, early misreadings (e.g., overweighting "10–11 narrowing," misinterpreting "sternal restriction") propagate forward to mutually inconsistent leaves, whereas the consistent path (2nd rib $\rightarrow$ large tuberosity $\rightarrow$ D) aligns with evidence.

**Mediation evidence.** Using early-window RGM as treatment, majority-path switching as mediator, and late-window CMHR as outcome, we find a significant indirect path (Sobel $z = 2.353$, $p = 0.0186$; $a \times b = 20.712$). The proportion mediated is approximately 225%, consistent with suppression where switching carries predictive signal even when the total marginal association is small. See Appendix A for assumptions, specification, and robustness.

**Takeaway.** Our preliminary investigation uncovers a critical and previously under-explored failure mode in multimodal clinical reasoning. We provide quantitative and qualitative evidence that reasoning instability, characterized by high entropy across independent generation paths, is not merely correlated with but is a direct precursor to hallucination. The analysis reveals a clear mechanism: early-stage stochasticity in interpreting evidence leads to divergent reasoning trajectories, which in turn propagate errors and result in globally inconsistent and factually incorrect conclusions. This finding directly motivates the design of our framework, MedMMV, which aims to shift the paradigm from uncontrollable generation to one of structured, verifiable, and stable reasoning.

## 4 METHODOLOGY

Given a clinical case with notes $\mathcal{T}$ and images $\mathcal{I}$, our proposed MedMMV produces a triplet $(\hat{y}, \hat{p}, \mathcal{E}_{\hat{p}})$: a diagnosis, its reasoning path, and the evidence graph that supports it. As shown in Figure 3, MedMMV proceeds in three stages: parallel path generation, supervised refinement, and uncertainty-aware selection, which are designed to enhance reliability and verifiability.

Figure 3: Overview of the MedMMV framework. Starting from patient notes and images, *Stage 1* produces diverse diagnostic paths through multi-round short rollouts. In *Stage 2*, paths are processed in parallel under the guidance of a Hallucination Detector and grounded in a multimodal evidence graph constructed by TextDoctor, ImageDoctor, and Researcher agents, enabling step-level verification. Finally, *Stage 3* aggregates candidates and selects the optimal diagnosis with a Combined Uncertainty scorer, yielding stable, evidence-supported, and interpretable reasoning.

## 4.1 STAGE 1: MULTI-ROUND SHORT ROLLOUTS GENERATION

Based on the insights from Section 3, we begin with uncertainty-aware hypothesis generation that seeds diversified paths. This mitigates the risk of committing to a single, potentially flawed, reasoning trajectory. We generate $k$ independent reasoning rollouts using a base MLLM. Each rollout produces a short, preliminary diagnostic path, denoted as $p_{\text{init}}$. This process yields a collection of initial paths, $\{p_{\text{init}}^{(1)}, p_{\text{init}}^{(2)}, \dots, p_{\text{init}}^{(k)}\}$. By starting with multi-round rollouts, we effectively sample the distribution of plausible initial diagnostic directions. This approach ensures that a wider range of possibilities is considered before committing to deeper, more resource-intensive analysis.

## 4.2 STAGE 2: PARALLEL, EVIDENCE-GROUNDED REASONING EXPANSION

The second stage is the cornerstone of MedMMV, where each initial path $p_{\text{init}}$ is subjected to a rigorous, parallel refinement process to ensure it is both factually grounded and logically coherent.

**Evidence graph construction.** To ground the reasoning process in verifiable facts, we first construct an evidence graph $\mathcal{E}$. This graph serves as a centralized, structured repository of all available evidence. We employ three specialized "doctor" agents to populate this graph: (1) **TextDoctor** parses the clinical notes $\mathcal{T}$ to extract structured entities such as symptoms, lab results, and patient history; (2) **ImageDoctor** analyzes medical images $\mathcal{I}$ to identify and describe abnormalities, including their location, size, and characteristics; (3) **WebSearch** queries external knowledge sources to find established medical relationships between the facts extracted by the other agents, providing supporting literature or clinical guidelines.

As illustrated in Figure 3, the graph $\mathcal{E}$ comprises nodes representing atomic facts and edges capturing the relations among them, with each edge annotated with its provenance. This structured, multi-agent approach makes factual connections explicit and auditable, providing a powerful mechanism for control and verification. For a detailed comparison that motivates our evidence-graph-based approach over full-context search, see Appendix E.3.

**Supervised path self-refinement.** Within each parallel track, the initial path $p_{\text{init}}$ is iteratively refined under the guidance of a Hallucination and Consistency Detector (HD Supervisor). The refinement proceeds through a tight verification-repair cycle. Claims in the current reasoning path are systematically fact-checked against the evidence graph $\mathcal{E}$. The supervisor flags any inconsistency, lack of support, or logical flaw. Based on this feedback, a targeted prompt is issued to the MLLM, instructing it to either re-examine the evidence or to revise or retract unsupported statements. The model then generates a revised reasoning step, which is immediately re-evaluated. This process continues until the path stabilizes. The final output of each module is a tuple $(p_{\text{final}}, \mathcal{E}_p)$, where $p_{\text{final}}$ denotes the refined and verified reasoning path and $\mathcal{E}_p \subseteq \mathcal{E}$ is the subgraph containing only the evidence directly supporting that path.

### 4.3 STAGE 3: DECISION AGGREGATION AND FINALIZATION

After the parallel expansion stage, we obtain a collection of refined candidate paths $\{(p_{\text{final}}^{(1)}, \mathcal{E}_p^{(1)}), \ldots, (p_{\text{final}}^{(m)}, \mathcal{E}_p^{(m)})\}$, where $m \leq k$. To select the single best diagnostic path, we introduce a Combined Uncertainty (CU) Score. This score evaluates each candidate as a weighted combination of evidence alignment, reasoning coherence, and repair history:

$$\text{CU}(p_{\text{final}}) = w_{\text{evidence}} \cdot S_{\text{evidence}}(p_{\text{final}}) + w_{\text{coherence}} \cdot S_{\text{coherence}}(p_{\text{final}}) - w_{\text{repair}} \cdot P_{\text{repair}}(p_{\text{final}})$$

Here, $S_{\text{evidence}}$ measures how well the path is supported by the evidence graph, and $S_{\text{coherence}}$ assesses its logical consistency, both derived from MLLM judgments. $P_{\text{repair}}$ denotes the number of auto-repair cycles during refinement. Since paths requiring fewer corrections are considered more reliable, the CU scorer naturally favors trajectories that remain stable and error-free. This ensures the chosen diagnosis is both robustly verified and demonstrably stable. Details on the CU score distribution and weight settings are provided in Appendix D.

## 5 EXPERIMENTS

Table 2: Overall accuracy results on six medical benchmarks.

| Category | Methods | Medical VQA | | | Medical QA | | |
|---|---|---|---|---|---|---|---|
| | | MedXpert-MM (T I) | MedFrameQA (T I) | PathVQA (T I) | MedXpert-Text (T) | MedQA (T) | MedMCQA (T) |
| CoT Baselines | GPT-5 | 67.90±2.19 | 46.95±2.87 | 67.91±3.10 | 32.40±2.12 | 99.07±0.72 | 35.34±3.50 |
| | GPT-4o | 37.28±1.98 | 42.62±3.13 | 37.03±3.08 | 19.49±1.84 | 94.81±1.47 | 32.39±3.42 |
| | GPT-oss-120B | 50.03±1.98 | 45.90±2.92 | 44.81±3.61 | 20.66±1.87 | 96.96±1.19 | 33.36±3.35 |
| | Claude-Sonnet-4 | 46.82±2.08 | 47.64±3.45 | 65.81±3.67 | 23.31±2.27 | 93.82±1.64 | 30.33±3.43 |
| | Qwen2.5-VL-7B | 10.50±5.48 | 20.31±7.01 | 49.21±4.33 | 10.67±8.00 | 48.99±3.47 | 26.52±3.00 |
| | Qwen2.5-VL-72B | 35.96±2.34 | 42.33±3.18 | 66.01±3.36 | 12.57±1.52 | 74.06±3.33 | 27.28±3.10 |
| Medical Agents | MDAgents | 44.21±3.57 | 39.36±3.45 | 64.66±3.25 | 17.48±2.58 | 81.08±2.62 | 31.02±3.76 |
| | ReConcile | 47.31±3.16 | 45.12±4.05 | 73.31±3.27 | 17.91±2.36 | 89.85±2.27 | 29.31±3.13 |
| | ColaCare | 33.20±3.45 | 39.06±3.42 | 72.19±3.15 | 14.96±2.31 | 88.93±2.63 | 29.05±3.27 |
| | MedAgent | 36.87±3.42 | 44.17±3.40 | 70.20±2.98 | 17.76±2.46 | 89.70±2.10 | 31.09±3.14 |
| MedMMV | GPT-5 | 73.58±1.87 | 59.67±1.76 | 68.25±2.08 | 39.46±1.08 | 99.15±3.00 | 39.62±2.69 |
| | GPT-4o | 60.24±2.33 | 53.08±2.45 | 39.46±3.53 | 26.31±3.31 | 94.62±3.53 | 38.64±3.13 |
| | GPT-oss-120B | 69.18±0.91 | 67.64±1.89 | 50.15±3.88 | 28.62±3.98 | 97.15±4.01 | 42.46±3.18 |
| | Claude-Sonnet-4 | 61.65±2.77 | 57.92±3.69 | 68.08±3.76 | 32.77±3.57 | 96.42±3.53 | 32.31±3.07 |
| | Qwen2.5-VL-7B | 35.35±2.34 | 40.15±6.01 | 51.15±4.29 | 18.62±6.54 | 56.08±4.78 | 36.03±6.23 |
| | Qwen2.5-VL-72B | 45.62±2.32 | 47.40±4.67 | 66.62±3.85 | 20.46±2.92 | 85.92±5.86 | 31.46±4.70 |

*Note:* (T): text modality; (I): image modality. Red shading highlights the top three methods for each dataset (darker red indicates higher performance). Accuracy is reported as mean (%) with standard deviation across multiple runs. MedMMV demonstrates consistent improvements across all tasks and backbones.

### 5.1 EXPERIMENTAL SETUPS

**Datasets.** We adopt six publicly available benchmarks that cover diverse modalities and task formats. For multimodal reasoning, we use **MedXpertQA-MM** (Zuo et al., 2025), **MedFrameQA** (Yu et al., 2025), and **PathVQA** (He et al., 2020). For text-only reasoning, we include **MedXpertQA-Text** (Zuo et al., 2025), **MedMCQA** (Pal et al., 2022), and **MedQA** (Jin et al., 2020). Following prior work, we evaluate on representative subsets (Appendix B).

**Baselines.** We benchmark against two categories: (1) *Direct prompting models*, including GPT-5 (Openai, 2025a), GPT-4o (Openai, 2024), GPT-oss-120B (Openai, 2025b), Claude-Sonnet-4 (Anthropic, 2025), Qwen2.5-VL-7B, and Qwen2.5-VL-72B (Bai et al., 2025a), prompted with zero-shot Chain-of-Thought (CoT); (2) *Agent-based systems*, including MDAgents (Kim et al., 2024), ReConcile (Chen et al., 2024), ColaCare (Wang et al., 2025b), and MedAgents (Tang et al., 2024). All agent baselines are instantiated with GPT-5 as the backbone MLLM for fair comparison.

**Implementation details.** MedMMV integrates modularly with the above MLLMs as executors. All models are accessed via official APIs. We set temperature to 0 for deterministic outputs, except GPT-5 which doesn't need temperature setting. Each trajectory allows up to three self-revision loops.

**Metrics.** *Accuracy* as the proportion of correct final choices. *TRUE / INFO (T, I)* as the averages over all questions of truthfulness score and informativeness score, and their product TRUE×INFO (joint quality, following (Lin et al., 2022; Wang et al., 2025a).), with raters $\mathcal{M} = \{\text{GPT-4o, Claude-Sonnet-4, Gemini}\}$.

## 5.2 RESULTS AND ANALYSIS

**Main results.** Table 2 shows that MedMMV consistently improves diagnostic accuracy over direct prompting and agent-based baselines. For instance, with GPT-5 as the executor, MedMMV increases accuracy from 67.9% to 73.6% (+5.7%) on MedXpert-MM and from 32.4% to 39.5% (+7.1%) on MedXpert-Text. Gains are most pronounced on complex multimodal benchmarks like MedFrameQA, where accuracy improves by 12.7%. Because each MedMMV variant uses the same backbone as its paired CoT baseline, these improvements are directly attributable to our controllable, multi-agent workflow rather than superior model capacity.

**Hallucination analysis.** Table 3 reports reliability-oriented metrics. We observe two consistent patterns. First, MedMMV yields higher truthfulness without sacrificing informativeness. With GPT-5, TRUE increases from 4.17 → 4.26 (MedXpert-MM) and from 3.68 → 4.22 (MedXpert-Text). This translates into joint quality (T×I) improvements of +10.9% and +10.2%. This shows that the workflow reduces hallucination without sacrificing information density. Second, baseline MLLMs and medical agents often achieve high INFO yet lag in TRUE. Across datasets, INFO remains stable ($\approx$ 4.5–4.7), showing gains come from more accurate, not just more verbose reasoning.

Table 3: Truthfulness (T), informativeness (I), and joint quality (T×I) on benchmarks.

| Category | Methods | Medical VQA | | | | | | | | | Medical QA | | | | | | | | |
| | | MedXpert-MM (T I) | | | MedFrameQA (T I) | | | PathVQA (T I) | | | MedXpert-Text (T) | | | MedQA (T) | | | MedMCQA (T) | | |
| | | T | I | T×I | T | I | T×I | T | I | T×I | T | I | T×I | T | I | T×I | T | I | T×I |
| CoT Baselines | GPT-5 | 4.17 | 4.78 | 72.03 | 3.44 | 4.57 | 65.84 | 3.51 | 4.59 | 68.00 | 3.68 | 4.70 | 71.36 | 4.92 | 4.98 | 98.30 | 2.67 | 4.56 | 51.00 |
| | GPT-4o | 3.31 | 4.57 | 56.47 | 3.23 | 4.24 | 57.40 | 3.51 | 3.54 | 58.67 | 2.82 | 4.32 | 51.98 | 4.72 | 4.90 | 93.82 | 2.56 | 4.51 | 48.53 |
| | GPT-oss-120B | 3.58 | 4.65 | 58.82 | 3.54 | 4.61 | 68.10 | 3.10 | 3.82 | 52.34 | 2.97 | 4.30 | 55.57 | 4.87 | 4.97 | 97.18 | 2.55 | 4.44 | 47.69 |
| | Claude-Sonnet-4 | 3.76 | 4.69 | 63.37 | 3.11 | 4.42 | 58.17 | 3.18 | 4.54 | 61.26 | 2.99 | 4.46 | 56.23 | 4.64 | 4.90 | 92.15 | 2.43 | 4.48 | 45.57 |
| | Qwen2.5-VL-7B | 1.68 | 2.53 | 18.22 | 2.12 | 2.23 | 24.44 | 2.74 | 3.08 | 39.14 | 1.20 | 2.10 | 11.15 | 2.81 | 2.65 | 36.12 | 1.79 | 2.50 | 21.03 |
| | Qwen2.5-VL-72B | 2.61 | 4.18 | 46.86 | 2.81 | 4.26 | 51.27 | 3.44 | 4.62 | 63.28 | 2.18 | 3.76 | 34.95 | 3.78 | 4.19 | 69.26 | 2.36 | 3.80 | 38.99 |
| Medical Agents | MDAgents | 2.11 | 1.84 | 17.19 | 3.08 | 4.50 | 58.32 | 3.30 | 4.60 | 65.19 | 2.08 | 3.78 | 44.32 | 4.33 | 4.71 | 85.41 | 2.31 | 4.00 | 45.23 |
| | ReConcile | 3.13 | 4.21 | 56.63 | 3.43 | 4.48 | 65.11 | 3.99 | 4.12 | 70.88 | 2.17 | 3.68 | 32.76 | 4.60 | 4.86 | 90.38 | 2.73 | 3.86 | 42.51 |
| | ColaCare | 2.72 | 4.09 | 48.44 | 3.39 | 4.52 | 52.27 | 3.49 | 4.19 | 72.10 | 2.39 | 3.63 | 35.27 | 4.64 | 4.79 | 90.16 | 2.61 | 3.70 | 37.82 |
| | MedAgent | 3.41 | 4.62 | 52.89 | 3.37 | 4.55 | 60.50 | 3.41 | 4.33 | 70.63 | 2.58 | 3.81 | 38.22 | 4.59 | 4.77 | 88.61 | 2.56 | 3.84 | 46.03 |
| MedMMV | GPT-5 | 4.26 | 4.73 | 82.89 | 4.00 | 4.42 | 74.22 | 3.74 | 4.02 | 65.36 | 4.22 | 4.70 | 81.53 | 4.94 | 4.82 | 95.36 | 2.97 | 4.57 | 56.40 |
| | GPT-4o | 4.20 | 4.42 | 76.70 | 3.72 | 3.90 | 60.00 | 3.61 | 3.76 | 59.58 | 3.55 | 4.05 | 60.22 | 4.95 | 4.95 | 97.96 | 2.58 | 4.93 | 49.02 |
| | GPT-oss-120B | 4.01 | 4.84 | 80.95 | 4.27 | 4.77 | 74.71 | 4.03 | 3.57 | 61.96 | 3.63 | 4.77 | 69.71 | 4.95 | 4.98 | 98.73 | 2.94 | 4.31 | 50.94 |
| | Claude-Sonnet-4 | 3.81 | 4.16 | 67.41 | 3.54 | 3.98 | 57.87 | 3.78 | 3.18 | 68.55 | 3.65 | 4.57 | 69.36 | 4.91 | 4.95 | 97.54 | 2.47 | 4.25 | 49.82 |
| | Qwen2.5-VL-7B | 2.59 | 2.59 | 30.06 | 3.33 | 3.30 | 45.76 | 3.27 | 3.52 | 46.02 | 2.28 | 3.08 | 20.42 | 3.70 | 3.72 | 47.85 | 2.13 | 2.72 | 23.71 |
| | Qwen2.5-VL-72B | 3.12 | 4.35 | 54.80 | 3.42 | 4.22 | 58.80 | 3.72 | 4.40 | 69.47 | 2.60 | 3.10 | 39.29 | 4.32 | 4.03 | 76.71 | 2.55 | 3.91 | 41.02 |

*Note:* (T): text modality; (I): image modality. Red shading marks the top three methods per dataset (darker red indicates higher performance). T: Truthfulness (1–5); I: Informativeness (1–5); T×I: normalized product in $[0, 100]$, computed as $(T \times I)/25 \times 100$. Scores are averaged over three independent LLM judges (GPT-5, Claude-Sonnet-4, Gemini), which helps mitigate single-model bias and prevents inflation from models rating their own outputs.

**Physician evaluation.** To complement automated metrics, we conducted a rigorous human evaluation to assess the clinical quality of generated responses. We recruited 27 licensed physicians, with specializations matching our nine medical categories, to perform a blind, head-to-head comparison of responses from MedMMV and CoT (both with GPT-5). Two to three physicians independently rated each response on 5-point scales for Truthfulness and Informativeness. As summarized in Table 4, the results demonstrate MedMMV's substantial advantage. Physician ratings confirm MedMMV's reliability. Averaged across all systems, MedMMV achieves a Truthfulness score of 4.36 versus 3.49 for CoT (+0.87), and a joint quality T×I score of 69.0 versus 58.7 (+10.3). The improvement is particularly notable in complex cases like cardiovascular, where MedMMV improves truthfulness from 3.15 to 4.28.

**Ablation study.** Table 5 reports the key ablations[1]. Replacing the CU Scorer with random selection reduces Accuracy by ∼11% on MedXpert-MM (∼6% on Text) and decreases TRUE×INFO by ∼13%, showing that uncertainty-aware aggregation is the dominant driver of end quality. Within Stage 2, disabling the self-feedback hallucination detector lowers Accuracy by ∼8% (MM) and ∼5% (Text), while removing path expansion results in losses of ∼5% (MM) and ∼3% (Text). These findings highlight hallucination control and parallel expansion as the most influential reasoning mechanisms. By contrast, removing specialist agents leads to only modest changes in Accuracy (1–4%). This suggests that modern MLLMs already perform a degree of implicit task decomposition.

---

[1]Abaltion study on the other four datasets can be referred in Appendix G.2

Table 4: Physician evaluation on the MedXpert-MM dataset by body system. **Note:** T: Truthfulness score (1–5); I: Informativeness score (1–5); T×I: Joint quality, normalized as $(T \times I)/25 \times 100$. Each body system includes 20 doctor-rated samples; the full per-doctor breakdown is in Appendix J.3.

| Body System | MedMMV | | | CoT | | |
|---|---|---|---|---|---|---|
| | **T** | **I** | **T×I** | **T** | **I** | **T×I** |
| Skeletal | $4.42_{\pm 0.34}$ | $4.13_{\pm 0.40}$ | $73.26_{\pm 12.87}$ | $4.11_{\pm 0.17}$ | $4.47_{\pm 0.38}$ | $73.39_{\pm 6.12}$ |
| Reproductive | $4.75_{\pm 0.23}$ | $4.00_{\pm 0.01}$ | $76.17_{\pm 3.60}$ | $3.60_{\pm 0.40}$ | $4.66_{\pm 0.27}$ | $67.23_{\pm 11.38}$ |
| Cardiovascular | $4.28_{\pm 0.46}$ | $3.97_{\pm 0.26}$ | $68.28_{\pm 11.95}$ | $3.15_{\pm 0.58}$ | $4.02_{\pm 0.33}$ | $51.12_{\pm 13.13}$ |
| Lymphatic | $4.23_{\pm 0.45}$ | $4.18_{\pm 0.39}$ | $71.28_{\pm 13.83}$ | $3.87_{\pm 0.60}$ | $3.98_{\pm 0.67}$ | $62.69_{\pm 20.83}$ |
| Nervous | $4.27_{\pm 0.55}$ | $3.68_{\pm 0.30}$ | $63.31_{\pm 13.41}$ | $3.18_{\pm 0.42}$ | $4.23_{\pm 0.63}$ | $54.59_{\pm 15.35}$ |
| Digestive | $4.19_{\pm 0.64}$ | $3.69_{\pm 0.30}$ | $62.46_{\pm 14.07}$ | $3.32_{\pm 0.20}$ | $4.03_{\pm 0.44}$ | $53.56_{\pm 7.16}$ |
| Urinary | $4.45_{\pm 0.43}$ | $4.03_{\pm 0.33}$ | $72.05_{\pm 11.95}$ | $3.73_{\pm 0.70}$ | $4.07_{\pm 0.35}$ | $61.37_{\pm 16.28}$ |
| Endocrine | $4.22_{\pm 0.29}$ | $3.63_{\pm 0.12}$ | $61.31_{\pm 5.22}$ | $3.02_{\pm 0.70}$ | $3.95_{\pm 0.22}$ | $47.35_{\pm 9.07}$ |
| Integumentary | $4.43_{\pm 0.42}$ | $4.11_{\pm 0.29}$ | $73.07_{\pm 11.04}$ | $3.44_{\pm 0.09}$ | $4.14_{\pm 0.40}$ | $57.03_{\pm 5.82}$ |
| **Average** | $\textbf{4.36}_{\pm 0.17}$ | $\textbf{3.94}_{\pm 0.20}$ | $\textbf{69.02}_{\pm 5.12}$ | $3.49_{\pm 0.34}$ | $4.17_{\pm 0.23}$ | $58.70_{\pm 7.77}$ |

The core advantage of our system therefore does not stem from the raw effect of individual agents, but from the controlled guidance of the *evidence graph*, which integrates signals from multiple modalities into a coherent and verifiable reasoning trajectory. To illustrate MedMMV's behavior, we provide case studies in Appendix F.1 and Figure 6. They show how the system avoids premature commitment in ambiguous scenarios and actively corrects hallucinated evidence.

Table 5: Ablation study on MedXpert-MM (multimodal) and MedXpert-Text (text-only). **Note:** Acc: Accuracy (%); T: Truthfulness (1–5); I: Informativeness (1–5); T×I: normalized product in [0,100], computed as $(T \times I)/25 \times 100$. ImageDoctor ablations are not applicable (–) for text-only tasks. Bold indicates full MedMMV with best performance.

| Category | Methods | MedXpert-MM (T)(I) | | | | MedXpert-Text (I) | | | |
|---|---|---|---|---|---|---|---|---|---|
| | | Acc | T | I | T×I | Acc | T | I | T×I |
| Full model | **MedMMV (GPT-5)** | **67.65** | **4.17** | **4.78** | **72.03** | **32.00** | **3.68** | **4.70** | **71.36** |
| Core framework | w/o Path Expansion with multi-round short CoT | 62.41 | 3.84 | 4.45 | 66.18 | 28.73 | 3.35 | 4.38 | 65.92 |
| | w/o Self-feedback halluc. detector | 59.23 | 3.52 | 4.51 | 64.07 | 27.45 | 3.18 | 4.42 | 63.74 |
| Expert agents | w/o TextDoctor agent | 63.78 | 3.91 | 4.52 | 67.85 | 29.34 | 3.42 | 4.48 | 67.21 |
| | w/o ImageDoctor agent | 65.12 | 3.98 | 4.63 | 69.47 | - | - | - | - |
| | w/o WebSearch agent | 66.29 | 4.08 | 4.71 | 70.84 | 30.87 | 3.61 | 4.65 | 70.15 |
| Decision strategy | Random decision instead of CU scoring | 56.84 | 3.12 | 4.19 | 58.91 | 26.15 | 2.89 | 4.08 | 57.43 |

## 6 DISCUSSION AND CONCLUSION

**Limitations.** First, the framework's deliberative, multi-path reasoning process inherently incurs greater computational and latency costs than single-pass inference models. In a high-stakes domain such as clinical medicine, where the cost of an error is exceptionally high, we argue this trade-off is not only justified but necessary for safe deployment. Nonetheless, this computational demand may limit its immediate applicability in time-critical scenarios, pointing toward initial use cases in non-emergent case analysis or second-opinion generation. Second, our evaluation is confined to benchmarks that may not fully replicate the dynamic and interactive nature of clinical practice. Finally, the integrity of MedMMV's reasoning is dependent on the accuracy of its initial evidence graph. Errors introduced by the specialist agents during this foundational step can propagate through the reasoning process, and the current architecture does not include a mechanism for correcting this graph post-generation, representing a potential vulnerability.

**Future work.** Future research could focus on extending MedMMV to longitudinal and interactive clinical settings, better reflecting real-world decision workflows. To fully leverage the framework's verifiability, we also plan to investigate human-in-the-loop interfaces that present the evidence graphs and competing reasoning paths to clinicians. In addition, we are exploring lightweight refinement and aggregation strategies to reduce computational overhead while maintaining reliability.

**Conclusion.** This work introduces MedMMV, a controllable multimodal multi-agent framework designed for reliable and verifiable clinical reasoning. By sampling diverse reasoning paths, verifying claims against a multimodal evidence graph, and selecting the most robust path via an uncertainty-aware scorer, MedMMV directly mitigates the failure mode where reasoning instability leads to

hallucination. Across six benchmarks, MedMMV improves both accuracy and reliability, with the largest gains on complex multimodal tasks where evidence selection and grounding are most critical. Our work underscores the importance of process-level control and offers a promising direction toward building safer and more auditable AI systems for healthcare.

## ETHICS STATEMENT

This work investigates the reliability of medical reasoning agents but is not intended for direct clinical deployment. All datasets are public and de-identified, and no patient data was collected. Physician evaluations were voluntary and confined to blinded response ratings. While MedMMV reduces hallucination, errors may still occur, and safe application requires regulatory approval, human oversight, and further validation in real-world settings. Biases in underlying MLLMs or benchmarks may propagate into outputs, and computational overhead may constrain deployment. We emphasize that MedMMV is a research prototype, aiming to advance responsible and auditable medical AI.

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

## A    PRELIMINARY ANALYSIS DETAILS

**Base metrics.**    For each question $q$ with $k_q$ generations, let $c_{q,i} \in \mathcal{O}_q$ be the option at round $i$. The empirical prior before round $i \geq 2$ is

$$\mathbf{p}_{q,i-1}(o) = \tfrac{1}{i-1} \sum_{t=1}^{i-1} \mathbf{1}\{c_{q,t} = o\}, \quad o \in \mathcal{O}_q.$$

Define the prior entropy $H_{q,i-1} = -\sum_{o \in \mathcal{O}_q} \mathbf{p}_{q,i-1}(o) \log \mathbf{p}_{q,i-1}(o)$ and per-round cross-modal hallucination $\mathrm{CMHR}_{q,i} \in [0, 100]$ from Eq. (4) in the main text. We exclude rounds $i = 1$ (no prior distribution), questions with final $H = 0$ (single-choice degeneracy), and rounds with missing scores.

**Common preprocessing.**    (i) Compute $(H_{q,i-1}, \mathrm{CMHR}_{q,i})$ for all rounds $i \geq 2$. (ii) Define **High–CMHR** as $\mathrm{CMHR}_{q,i} \geq \tau$, default $\tau$ at the global 75th percentile (threshold robustness below). (iii) For each question, compute **churn** $S_q = \sum_{i=2}^{k_q} \not\Vdash \{m_{q,i} \neq m_{q,i-1}\}$ where $m_{q,i}$ is the majority option under $\mathbf{p}_{q,i}$ (ties broken arbitrarily).

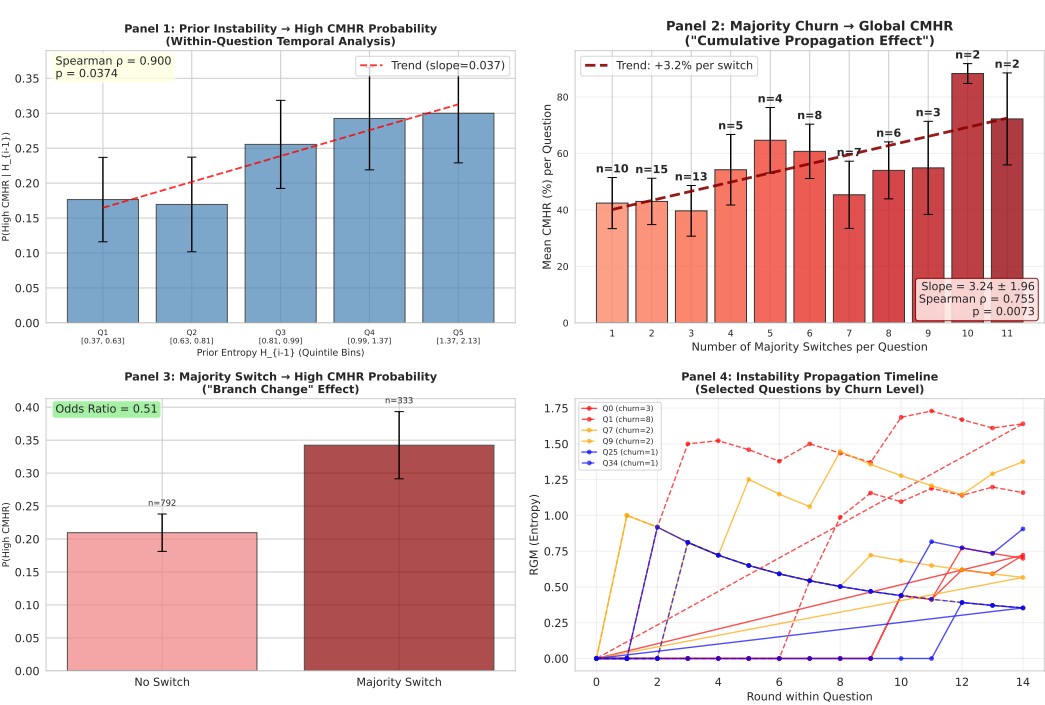

Figure 4: Instability breeds hallucination. *Panels 1–4:* Prior instability strongly predicts hallucination risk. (1) Higher entropy at step $i - 1$ increases the probability of high CMHR at step $i$. (2) Majority-path churn accumulates globally and correlates with higher mean CMHR. (3) A single majority-path switch produces an immediate +13.3% increase in hallucination probability. (4) Timelines show that high-churn questions propagate instability across reasoning steps, while low-churn questions remain stable.

### A.1    PANEL 1: PRIOR INSTABILITY → HIGH CMHR (WITHIN-QUESTION INSTANTANEOUS EFFECT)

**Goal.** Assess whether higher prior instability predicts hallucination risk in the subsequent generation.
**Construction.**

(1) Bin $H_{q,i-1}$ into quintiles (Q1–Q5) across all $(q, i)$.

(2) For each bin $b$, estimate $\widehat{p}_b = \Pr(\text{High-CMHR}_{q,i} = 1 \mid H_{q,i-1} \in b)$.

(3) Plot $\widehat{p}_b$ with 95% CIs from clustered bootstrap (5,000 resamples clustered by question).

(4) Fit a monotone trend (OLS on bin midpoints) and report Spearman's $\rho$ with exact $p$-values.

**Note.** This is a within-question analysis using only prior entropy for the same item.

## A.2 Panel 2: Majority Churn → Global CMHR (Cumulative Propagation)

**Goal.** Examine whether local instability aggregates into global inconsistency across the trajectory.
**Construction.**

(1) For each question $q$, compute churn $S_q$ and global inconsistency $\overline{\text{CMHR}}_q = \frac{1}{k_q-1} \sum_{i=2}^{k_q} \text{CMHR}_{q,i}$.

(2) Plot $\overline{\text{CMHR}}_q$ against $S_q$ (grouped if sparse) with mean±SE bars.

(3) Estimate a robust OLS regression $\overline{\text{CMHR}}_q = \alpha + \gamma S_q + \varepsilon_q$ (HC1 standard errors) and report $\hat{\gamma}$.

(4) Report Spearman $\rho$ and $p$ for rank correlation.

## A.3 Panel 3: Majority Switch → High CMHR (Branch-Change Effect)

**Goal.** Translate entropy into a mechanism: test whether a branch switch increases hallucination risk within the same round.
**Construction.**

(1) Define a majority-switch indicator at round $i \geq 2$: $Z_{q,i} = \mathbf{1}\{m_{q,i-1} \neq c_{q,i}\}$.

(2) Form two groups: no-switch ($Z = 0$) vs. switch ($Z = 1$).

(3) Estimate $\Pr(\text{High-CMHR} \mid Z)$ with 95% clustered bootstrap confidence intervals.

(4) Report Fisher's exact test and the odds ratio (no-switch vs. switch).

**Optional.** Logit with question fixed effects: $\Pr(\text{High-CMHR}_{q,i} = 1) = \sigma(\alpha_q + \beta Z_{q,i})$; clustered SEs by question.

## A.4 Panel 4: Instability Propagation Timelines (Selected Questions)

**Goal.** Visualize how instability accumulates over rounds for high- vs. low-churn items and link to the case tree.
**Construction.**

(1) Select representative questions at different churn levels (e.g., $S \in \{1, 2, 3, 8, 11\}$).

(2) For each selected $q$, plot round-wise cumulative entropy $H_{q,i}$ (or $H_{q,i-1}$) over $i$; optionally smooth with a 3-point moving average.

(3) Use a common y-axis and distinct line styles; annotate switches to show where branch changes occur.

**Link.** Mark the high-entropy fork (Step 2) in the case-study tree; align rising segments in timelines with the error flow (misfocus/mis-mapping → wrong leaf).

**Plotting and inference defaults.** CIs: 95% via clustered bootstrap (5,000 resamples, cluster=question) unless noted.

Nonparametrics: Spearman $\rho$ with exact (or large-sample) $p$.

Trends: OLS with HC1 robust SE; shaded band denotes 95% CI.

Exclusions: rounds $i = 1$, items with final $H = 0$, and missing CMHR.

These analyses mirror real-world diagnostic uncertainty: early misinterpretation of evidence (instability) makes clinicians more likely to shift hypotheses, which increases the risk of introducing unsupported findings.

**Robustness variants.** High–CMHR thresholds: $\tau$ at 70/75/80th percentiles or $z$-score $> 0.5$.

Alternative binning (quintiles vs. deciles) yields qualitatively unchanged results.

Panel 1/3: fixed-effects logit; Panel 2: panel-OLS with question fixed effects.

Alternative switch definition: tie-aware majority and "top-2 margin $< \delta$" treated as *uncertain* (robust to $\delta \in [0.05, 0.15]$).

**Mediation analysis (early instability $\Rightarrow$ switching $\Rightarrow$ late hallucination).** *Causal chain and identification.* We posit the directed chain

$$\mathrm{RGM}^{\mathrm{early}}\,(\,T\,) \;\longrightarrow\; \mathrm{Switch}\,(\,M\,) \;\longrightarrow\; \mathrm{CMHR}^{\mathrm{late}}\,(\,Y\,),$$

interpreted under: (A) *Temporal ordering*: $T$ from early rounds, $M$ from a middle window, $Y$ from late rounds; (B) *Sequential ignorability*: conditional on controls $X$ (item type, difficulty proxies, and question fixed effects), there are no unmeasured confounders of $T \to M$ and $M \to Y$ affected by $T$; (C) *Consistency & positivity*.
*Specification.* Let $T_i = \mathrm{RGM}^{\mathrm{early}}$, $M_i = \mathrm{Switch}$ (count or rate in the mediator window), $Y_i = \mathrm{CMHR}^{\mathrm{late}}$. We estimate

$$\mathrm{Mediator:} \quad M_i = \alpha + a\,T_i + \gamma^\top X_i + \varepsilon_i,$$
$$\mathrm{Outcome:} \quad Y_i = \alpha' + c\,T_i + b\,M_i + \delta^\top X_i + \varepsilon_i',$$

with heteroskedasticity-robust SEs clustered at the question level; fixed-effects variants and nonparametric bootstrap (for $a$, $b$, and $ab$) are also considered.
*Estimates.*

$$\beta_a = 5.319\pm1.360\,(p < 10^{-4}), \quad \beta_b = 3.894\pm3.087\,(p = 0.0158), \quad \beta_c = 9.201\pm18.927\,(p = 0.3438).$$

The indirect effect is significant (Sobel $z=2.353$, $p=0.0186$; $a \times b = 20.712$), with a proportion mediated of $\sim$225%, consistent with a suppression pattern where the mediator transmits predictive signal despite a weak marginal $T \to Y$ association.
*Optional DAG.*

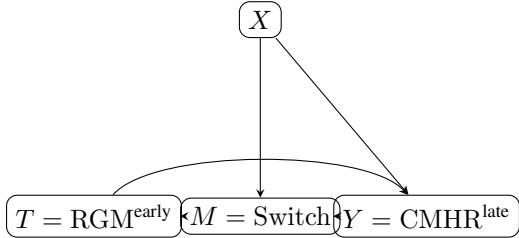

Figure 5: Assumed mediation structure with controls $X$.

## B  DATASET STATISTICS

To better evaluate the performance of our model across diverse medical reasoning tasks, we collect and use several widely adopted datasets from both visual question answering (VQA) and text-based question answering (QA) domains. Table 6 summarizes the dataset statistics. Due to computational resource constraints, we report results on representative subsets for evaluation while ensuring coverage across different modalities (image + text vs. text-only). We select subsets by stratified sampling to preserve the diversity of question types and difficulty levels.

## C  ALGORITHMIC DETAILS

The complete MedMMV workflow is formalized in Algorithm 1. The procedure begins with Stage 1 (Hypothesis Generation), where the MLLM produces $k$ short diagnostic rollouts from clinical notes and images, yielding a diverse set of initial reasoning paths $\mathcal{P}_{\mathrm{init}}$.

Table 6: Statistics of the six medical QA/VQA benchmarks used in our study. Each dataset includes the original size and the representative subset size used for evaluation, chosen to balance coverage and computational feasibility.

| Dataset | Task | Modality | Original size | Subset size |
|---|---|---|---|---|
| MedXpertQA-MM | VQA | Image + Text | 10,868 | 238 |
| MedFrameQA | VQA | Image + Text | 2,851 | 200 |
| PathVQA | VQA | Image + Text | 32,799 | 200 |
| MedXpertQA-Text | QA | Text | 26,675 | 200 |
| MedMCQA | QA | Text | 193,155 | 200 |
| MedQA | QA | Text | 12,723 | 200 |

In Stage 2 (Parallel Evidence-Grounded Refinement), we first construct a unified evidence graph $\mathcal{E}$ using the three specialist modules: `TextDoctor` for structured entities from notes, `ImageDoctor` for image-derived findings, and `WebSearch` for external clinical knowledge. Each initial path is then refined in parallel under the supervision of the Hallucination and Consistency Detector (HD Supervisor). At each iteration, reasoning claims are fact-checked against $\mathcal{E}$; if inconsistencies or unsupported statements are detected, targeted feedback triggers the `AutoRepair` process, revising the path until convergence. The outcome is a set of refined paths paired with their supporting subgraphs $\mathcal{C}_{\text{refined}}$.

Finally, Stage 3 (Decision Aggregation) selects the optimal diagnosis. Each candidate path is evaluated by the Combined Uncertainty (CU) Scorer, which integrates evidence alignment, coherence, and repair cost into a single reliability score. The highest-scoring path $(\hat{p}, \mathcal{E}_{\hat{p}})$ is chosen, and the final diagnosis $\hat{y}$ is extracted from it. This ensures that the reported output is not only accurate but also explicitly verified against multimodal evidence.

## D  COMBINED UNCERTAINTY (CU) SCORE

The CU score for a given final path $p_{\text{final}}$ is calculated as:

$$\text{CU}(p_{\text{final}}) = w_{\text{evidence}} \cdot S_{\text{evidence}}(p_{\text{final}}) + w_{\text{coherence}} \cdot S_{\text{coherence}}(p_{\text{final}}) - w_{\text{repair}} \cdot P_{\text{repair}}(p_{\text{final}})$$

where $S_{\text{evidence}}$ measures the proportion of claims in the path successfully verified against the evidence graph $\mathcal{E}_p$, $S_{\text{coherence}}$ is a score assigned by an MLLM evaluator judging the logical flow, and $P_{\text{repair}}$ is the number of corrections made during the Stage 2 refinement. In our experiments, we set all weights $w_i$ to 1 for simplicity, although they can in principle be rescaled. To better illustrate the distribution of these four metrics, we report summary statistics for a representative model in Table 7.

## E  PROMPT AND OUTPUT EXAMPLE OF AGENTS

To ensure transparency and reproducibility, we provide representative prompts and output examples of the core agents in MedMMV. These include the *Text Doctor*, which extracts structured findings from clinical notes, the *Image Doctor*, which objectively describes medical imaging features, and the *Hallucination Detector*, which flags fabricated or unsupported reasoning. We also illustrate the *Researcher* module's evidence-graph–based search compared to whole-context search, highlighting how structured verification enables more precise and traceable reasoning. Together, these examples demonstrate how MedMMV operationalizes controllable reasoning through standardized agent behaviors.

**Algorithm 1:** The MedMMV Framework Algorithm

**Input:** Clinical notes $\mathcal{T}$; Medical images $\mathcal{I}$; Number of rollouts $k$.
**Output:** Final diagnosis $\hat{y}$, diagnostic text $\hat{p}$, evidence graph $\mathcal{E}_{\hat{p}}$.

```
// Stage 1: Generation of Diverse Initial Hypotheses
```
$\mathcal{P}_{\text{init}} \leftarrow \emptyset$;
**for** $i \leftarrow 1$ **to** $k$ **do**
  $p_{\text{init}}^{(i)} \leftarrow \text{MLLM.generate\_rollout}(\mathcal{T}, \mathcal{I})$;
  $\mathcal{P}_{\text{init}} \leftarrow \mathcal{P}_{\text{init}} \cup \{p_{\text{init}}^{(i)}\}$;

```
// Stage 2: Parallel, Evidence-Grounded Reasoning Expansion
```
$\mathcal{E} \leftarrow \text{BuildEvidenceGraph}(\text{TextDoctor}(\mathcal{T}), \text{ImageDoctor}(\mathcal{I}), \text{WebSearch}(\ldots))$;
$\mathcal{C}_{\text{refined}} \leftarrow \emptyset$;
```
// Process each initial path in parallel
```
**for** $p_{init}$ **in parallel** from $\mathcal{P}_{init}$ **do**
  $p_{\text{current}} \leftarrow p_{\text{init}}$;
  **while** *not converged* **do**
    feedback $\leftarrow \text{HDSupervisor}(\text{FactCheck}(p_{\text{current}}, \mathcal{E}))$;
    **if** *feedback is empty* **then**
      break;
    $p_{\text{current}} \leftarrow \text{AutoRepair}(p_{\text{current}}, \text{feedback})$;

  $p_{\text{final}} \leftarrow p_{\text{current}}$;
  $\mathcal{E}_p \leftarrow \text{ExtractRelevantSubgraph}(\mathcal{E}, p_{\text{final}})$;
  Add $(p_{\text{final}}, \mathcal{E}_p)$ to $\mathcal{C}_{\text{refined}}$;

```
// Stage 3: Aggregation and Final Selection
```
$(\hat{p}, \mathcal{E}_{\hat{p}}) \leftarrow \arg\max_{(p_j, \mathcal{E}_j) \in \mathcal{C}_{\text{refined}}} \text{CUScorer}(p_j, \mathcal{E}_j)$;
$\hat{y} \leftarrow \text{ExtractDiagnosis}(\hat{p})$;

**return** $\hat{y}, \hat{p}, \mathcal{E}_{\hat{p}}$;

Table 7: Distribution of scores for GPT-5 on the CU evaluation metrics.

| Statistic | $S_{\text{evidence}}$ | $S_{\text{coherence}}$ | $P_{\text{repair}}$ | CU |
|-----------|-----------|-----------|-----------|-----------|
| Mean | 0.78 | 0.72 | 0.31 | 1.19 |
| Std. Dev. | 0.12 | 0.15 | 0.09 | 0.18 |
| Min | 0.50 | 0.40 | 0.10 | 0.80 |
| Max | 0.95 | 0.95 | 0.50 | 1.50 |

### E.1 TEXT DOCTOR PROMPT

> **Expert Agent Prompt**
>
> PROMPT = """
> You are a medical domain expert analyzing clinical text data. Your task is to identify key symptoms, conditions, and clinical findings from the patient information provided.
> CLINICAL TEXT: {**Clinical text**}
> Consider this may be a medical exam question with a specific diagnosis in mind. Extract family medical history, past medical history, relevant symptoms, medical conditions, and clinical findings carefully. Organize findings by body system or symptom category. Note any significant patient history that could impact diagnosis.
> FORMAT YOUR RESPONSE EXACTLY AS FOLLOWS:
> FAMILY MEDICAL HISTORY:
> - [ family condition 1 ]
> PAST MEDICAL HISTORY: - [ past condition 1 ]
> EXTRACTED SYMPTOMS:
> - [symptom 1]
> CLINICAL FINDINGS:
> [Body System 1]:
> - [finding 1]
> [Body System 2]:
> - [finding 1]
> Ensure your output follows this exact format for automated processing. """

### E.2 IMAGE DOCTOR PROMPT

> **Expert Agent Prompt**
>
> PROMPT = """
> Your task is to carefully and objectively describe the visible imaging findings, without making diagnostic judgments or clinical assumptions.
> PATIENT INFORMATION: {**patient info**}
> MEDICAL IMAGES: {**medical images**}
> Please identify and describe all visible abnormalities, focusing only on what is directly observable. Do not speculate or provide diagnoses.
> For each finding, describe the following:
> - Region or organ involved
> - Distribution: focal / diffuse / multifocal
> - Number: single / few / multiple / innumerable
> - Size: provide approximate range (e.g., 1–3 mm, <1 cm, >3 cm)
> - Density: solid / ground-glass / cavitary / mixed
> - Margins: well-defined / ill-defined
> - Laterality: unilateral / bilateral
> - Associated features: pleural effusion, lymphadenopathy, airway distortion, cavitation, etc. (only if visible)
> Be concise, objective, and specific. Avoid speculative language or uncertain modifiers (e.g., "may represent", "possibly"). Only report what is visually evident.
> FORMAT YOUR RESPONSE EXACTLY AS FOLLOWS:
> IMAGE FINDINGS:
> - [Region/Organ]: [Concise structured description with above attributes] """

### E.3 RESEARCHER PSEUDOCODE AND COMPARING WITH WHOLE CONTEXT SEARCH

We adopt an evidence-graph approach instead of whole-context search, which often reduces retrieved text to undifferentiated facts and loses critical relationships among findings, interventions, and

outcomes. By explicitly modeling entities and typed links, and verifying with external literature, evidence graphs enable clinically meaningful connections, conflict detection, and traceable reasoning.

---

**Algorithm 2:** Medical Literature Web Search

---

**Input:** Evidence Graph object $evidencegraph$; Configuration $config$.
**Output:** Web search results with medical findings.

$evidencegraph \leftarrow$ GetObservations($state$);
**if** $evidencegraph = \emptyset$ **then**
    | **return** EmptySearchResult ();

$combinations \leftarrow$ GeneratePairs($evidencegraph$, max $= 8$);
$llm \leftarrow$ InitializeChatGPT($config$);
$findings \leftarrow [\,]$;
**for** $combo \in combinations$ **do**
    | $query \leftarrow$ BuildMedicalQuery($combo$);
    | $results \leftarrow$ WebSearchAPI($query$);
    | $context \leftarrow$ ExtractContent($results$);
    | $analysis \leftarrow llm$.Analyze($context$);
    | $findings$.append($combo, analysis$);

$merged \leftarrow$ FormatFindings($findings$);
**return** SearchState ($merged, combinations$);

---

### Whole Context Search

RESULTS = """
**Page 1**: Purpose To define the role of MRI in the diagnosis and management of Chance-type flexion distraction injury.
Results At MRI, combined bony and soft tissue injuries were more common than either bone or soft tissue damage alone, and occurred at the thoracolumbar junction primarily. Contiguous vertebral injury was seen in 18 cases, with non-contiguous injury in 7 cases. Posterior ligamentous complex disruption occurred in all cases. Extensive subcutaneous and para-spinal muscle oedema was seen in all cases extending over several segments. Horizontally orientated fractures of the posterior neural arches produced a distinctive MRI pattern -"Sandwich sign"- consisting of linear haemorrhage framed by marrow oedema. Extension of the fracture into the posterior vertebral body outline occurred in 3 cases, with fracture displacement into the canal. The posterior vertebral body height remained unchanged or increased in these 3 cases. **Conclusion:** The MR features of CTFD injuries are recognisable and reproducible, allowing a confident differentiation from "unstable burst injuries".
**Page 2**: **Context**: Arthroscopic reconstruction of ACL is an effective method to restore knee stability after ACL rupture. Postoperative septic arthritis (SA) is very uncommon while the incidence of serious complications range between 0.14% and 1.8%. Some of the devastating consequences of septic arthritis can encompass hyaline cartilage damage, arthrofibrosis, and in rare cases amputation. The purpose of this study was to evaluate the effect of gentamicin irrigation solutions as a process to restrain septic arthritis following arthroscopic ACL reconstruction. **Results**: Seven patients developed SA, four of whom were from SALINE group (2.2%) and three from Gentamycin group (0.23%). The incidence rate of SA after arthroscopic ACL reconstruction was significantly lower (P <0.05) when irrigated with gentamicin solution than merely with saline solution. **Conclusion:** Gentamicin irrigation solution has a preservative and protective effect against SA development following arthroscopic ACL reconstruction. We recommend evaluating this technique as a way in order to depreciate the prevalence of SA after ACL reconstruction. """

---

**Evidence Graph Search**

RESULTS = """
**[Fever + Pain]**: Fever and pain are often associated, with pain sometimes being a symptom of the underlying illness causing the fever, or the fever itself causing symptoms like headache or muscle aches. In children presenting with fever and limb pain, acute lymphoblastic leukemia (ALL) should be considered, especially if accompanied by fatigue, bruising, or bleeding. In infants and toddlers, ibuprofen may be more effective than acetaminophen for fever reduction and pain relief.
**[Fever + Wound drainage at the surgical site]**: Fever and wound drainage at the surgical site can indicate a surgical site infection (SSI), which may involve superficial or deep tissues and sometimes manifests with heat, redness, swelling, and purulent exudate. While fever is a common response to surgery, a persistent or high fever accompanied by wound drainage should be evaluated to rule out infection.
**[Fever + ruptured Achilles tendon with a 5 cm gap]**: While fever is not directly established as a common association of a ruptured Achilles tendon with a 5 cm gap, systemic diseases including infections can be related to Achilles tendon injuries. A ruptured Achilles tendon is often characterized by sudden pain, swelling, and impaired movement. A larger gap size in the ruptured tendon may predict lower patient-reported outcomes.
**[Fever + Elevated ESR of 29 mm/hr (normal range: 0)]**: Fever with an elevated ESR of 29 mm/hr (normal range: 0) can be associated with systemic or bone infections, heart conditions, rheumatic fever, severe skin infections, tuberculosis, autoimmune disorders, certain cancers like lymphoma or multiple myeloma, kidney or thyroid disease, anemia, pregnancy, diabetes mellitus, end-stage renal disease, heart disease, malignancy, allergic vasculitis...
**[Fever + Past Medical History: [Achilles tendon repair]]**: A fever following Achilles tendon repair could indicate a potential infection, warranting immediate contact with the referring physician. """

---

E.4 Hallucination Detector Prompt

---

**Expert Agent Prompt**

PROMPT = """
You are a medical hallucination detector. Your task is to identify any fabricated, imagined, or unsupported medical information in the analysis.
Original Clinical Text **original clinical text**
Analysis Type: {**analysis type**}
Analysis to Check {**analysis text**}
Please identify:
1. FABRICATED INFORMATION
2. UNSUPPORTED CLAIMS
3. CONTRADICTIONS
Format your response as:
HALLUCINATION DETECTED: [YES/NO]
FABRICATED INFORMATION: - [List any fabricated details, or "None detected"]
UNSUPPORTED CLAIMS: - [List any unsupported medical claims, or "None detected"]
CONTRADICTIONS: - [List any contradictions, or "None detected"]
CONFIDENCE SCORE: [0-100]% (How confident you are in this hallucination assessment)
RECOMMENDATION: Provide specific actionable recommendation
- choose from: ACCEPT AS IS, ACCEPT WITH CAUTION, REVISE REDUCE FABRICATION, REVISE STRENGTHEN EVIDENCE, REVISE REMOVE CONTRADICTIONS, REVISE COMPREHENSIVE, REJECT HIGH HALLUCINATION, REJECT UNRELIABLE
SPECIFIC ACTIONS: List 2 to 3 specific actions to improve the analysis.
SEVERITY: LOW/MEDIUM/HIGH """

---

The example JSON output of hallucination detector is as follows:

```
1188  {
1189    "analysis_name": "Image Doctor Analysis",
1190    "attempt": 1,
1191    "hallucination_detected": "YES",
1192    "recommendation": "REVISE_REDUCE_FABRICATION",
1193    "total_issues": 4,
1194    "severity": "MEDIUM",
1195    "improvement_guidance_applied": false,
1196    "improvement_instructions": [
1197      "Action: Base imaging interpretation only on actually visible findings in
            ↪  the image.",
1198      "Action: Eliminate unsupported claims about the number of lesions.",
1199      "Do not fabricate: Lesions vary, approximately 1-3 cm in diameter is not
1200         ↪  mentioned in the original clinical text.",
1201      "Unsupported claim to avoid: The description of multiple lesions is not
1202         ↪  explicitly supported by the original clinical text or visible in the
1203         ↪  image.",
1204    ],
1205    "previous_warnings_count": 0
1206  }
```

## F  CASE STUDY

To qualitatively demonstrate the capabilities of MedMMV, we present several case studies in the Appendix F.1 and Figure 6. We compare the reasoning process of our agent with that of a direct CoT. These case studies illustrate two important properties. First, in *ambiguous scenarios* where multiple diagnostic hypotheses are initially plausible, MedMMV expands paths in parallel and uses the evidence graph to explicitly weigh supporting vs. conflicting findings. This prevents premature commitment to a single hypothesis and allows the system to maintain clinically reasonable alternatives until sufficient evidence is gathered. These case studies illustrate two important properties. First, in *ambiguous scenarios* where multiple diagnostic hypotheses are initially plausible, MedMMV expands paths in parallel and uses the evidence graph to explicitly weigh supporting vs. conflicting findings. This prevents premature commitment to a single hypothesis and allows the system to maintain clinically reasonable alternatives until sufficient evidence is gathered. Second, when *errors arise*, such as fabricated or misinterpreted evidence in an early path, the hallucination detector flags the inconsistency and feeds it back to the corresponding reasoning block. This allows MedMMV to actively revise the erroneous evidence and update the evidence graph. The case studies highlight how MedMMV's controllable reasoning enables it to handle ambiguous clinical scenarios, correct its own errors, and ultimately lead to a more reliable and clinically sound diagnosis.

### F.1  TYPICAL REPAIRED EXAMPLE COMPARISON.

To further highlight the advantages of MedMMV over direct chain-of-thought (CoT) reasoning, we present a representative example where the baseline model fails while our approach successfully repairs the reasoning (Table 8). The case involves a patient with an infected re-ruptured Achilles tendon. Direct CoT incorrectly prioritizes immediate reconstruction, overlooking the contraindication posed by active infection. In contrast, MedMMV explicitly aligns textual, visual, and clinical evidence to recognize infection as a dominant factor, thereby deferring reconstruction in favor of thorough debridement. This example illustrates how MedMMV prevents reasoning shortcuts, repairs erroneous diagnostic chains, and produces more clinically faithful outputs by enforcing evidence alignment and highlighting bifurcation points during the reasoning process.

### F.2  EXAMPLE WITH INTERMEDIATE OUTPUT.

To better illustrate how MedMMV performs step-by-step reasoning, we provide a representative case study with intermediate outputs (Figure 6). The system begins with **Stage 1**, where multiple short diagnostic rollouts are generated by a junior diagnostician, each associated with an initial hypothesis and a preliminary fact check. In **Stage 2**, these candidate diagnostic paths are expanded in

Table 8: Comparing the Outputs of Direct CoT and MedMMV on GPT-oss-120B.

| **Question** |
| --- |
| A 36-year-old male patient presents with fever, pain, and wound drainage at the surgical site 4 months following an Achilles tendon repair. Figure A shows the clinical presentation. Laboratory testing reveals an elevated ESR of 29 mm/hr (normal range: 0-22 mm/hr). During the planned debridement and irrigation procedure, it is discovered that the Achilles tendon has re-ruptured with a 5 cm gap. Given these findings, what is the most appropriate next step in surgical management? **Options:** A: Primary repair of the re-ruptured Achilles tendon B: Repair of the re-ruptured Achilles tendon with a turndown procedure C: Repair of defect with flexor hallucis longus tendon transfer D: Debridement of necrotic and infected tendon tissue, with no attempt at reconstruction E: V-Y plasty of the re-ruptured Achilles tendon |

| **Direct CoT** | **MedMMV (ours)** |
| --- | --- |
| **Selected Answer: (C)** **Evidence cited:** – Large gap (5 cm) ⇒ requires FHL transfer immediately. – Infection can be managed simultaneously with reconstruction. **Issue:** Fails to recognize that active infection contraindicates immediate reconstruction. **Clinical Reasoning:** The infection and large gap necessitate both adequate debridement and effective reconstruction. Flexor hallucis longus tendon transfer is appropriate for large defects and aids in eliminating infected tissue while providing functional reconstruction. | **Selected Answer: (D)** **Corrected Evidence:** – Active infection + necrotic tissue ⇒ reconstruction must be deferred. – Primary repair (A), turndown (B), V-Y plasty (E) are contraindicated during infection. – FHL transfer (C) is valid for chronic >3 cm defects, but only after infection eradication. **Reasoning:** Immediate priority is thorough debridement of infected tissue to prepare for later reconstruction. **Clinical Reasoning:** Given these findings, at this stage the priority is infection control rather than reconstruction. Debridement, which involves debridement of necrotic and infected tissue without reconstruction, is a more conservative approach that addresses the infection first. This is crucial to prevent further complications and to create a healthier environment for any future reconstructive procedures. Therefore, the most appropriate next step in surgical management, given the active infection and necrotic tissue, is to focus on debridement. |

parallel through consultation with text-based doctors, image specialists, and researcher modules. Each expansion produces structured evidence nodes and cross-modal connections, resulting in evidence maps that make the reasoning process transparent. Finally, in **Stage 3**, the outputs from parallel paths are aggregated and scored. The system then selects the most consistent and well-supported reasoning chain as the final output. This case demonstrates how MedMMV integrates textual, visual, and clinical knowledge sources to arrive at a robust diagnosis while explicitly showing intermediate reasoning states.

# G ABLATION STUDY DETAILS

## G.1 ABLATION SETTINGS

This section provides precise description of the ablation settings presented in Table 5.

**Full model.** This is the complete MedMMV system as described in the main paper. It utilizes all stages. This configuration serves as the main baseline for all comparisons.

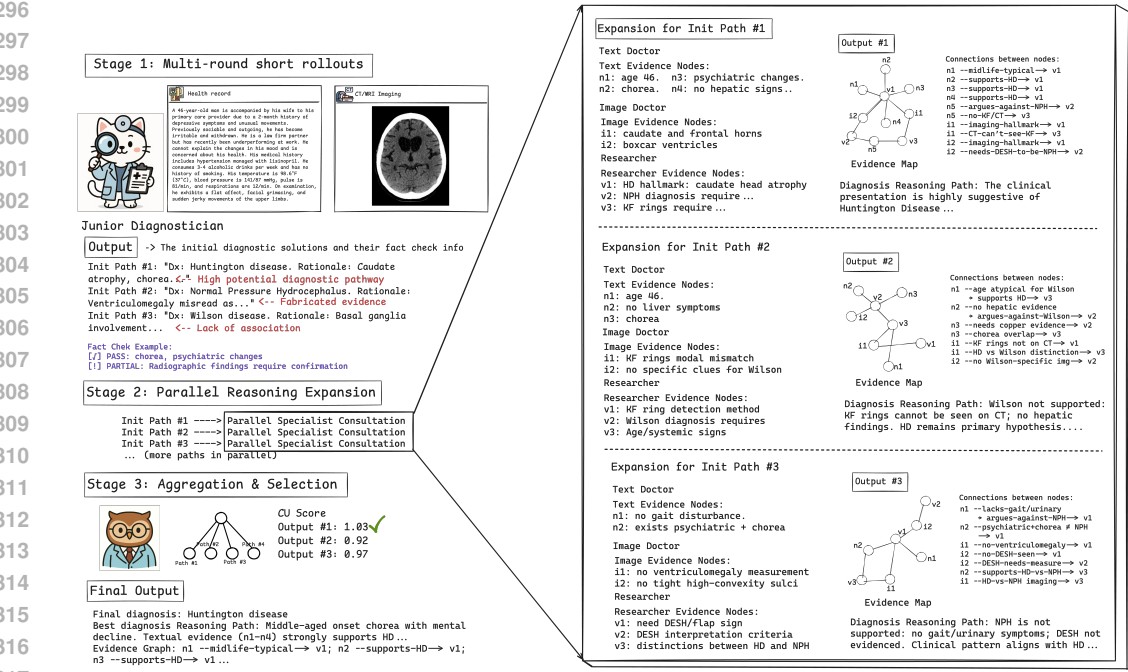

Figure 6: Case study example of MedMMV with intermediate outputs.

**w/o parallel reasoning expansion.** In this setting, we replace the core architecture of generating and expanding multiple parallel paths. Instead, the model employs a single, sequential reasoning process with evidence-grounded reasoning. The model iteratively refines its single line of reasoning, but does not explore the diverse hypothesis space that our parallel framework is designed to cover. This tests the benefit of concurrent exploration.

**w/o HD supervisor.** The Hallucination Detection (HD) Supervisor, a key component within each Reasoning Expansion Module, is deactivated. Consequently, the reasoning paths are generated without the internal self-correction loop. The Auto-Repair step, which relies on feedback from the supervisor to fix logical fallacies or factual errors, is effectively skipped. This ablation isolates the impact of our intra-path consistency and factuality checks.

**w/o TextDoctor agent.** The system is prohibited from invoking the TextDoctor agent during the Evidence Integration step of Stage 2. All reasoning must be based on evidence sourced from the original text input. This directly measures the contribution of structured text evidence extraction from clinical information.

**w/o ImageDoctor agent.** Similar to the TextDoctor ablation, the ImageDoctor agent is made unavailable. Forcing the system to rely solely on original medical image information rather than MedMMV's refined and grounded visual evidence. This configuration is only applicable to the multimodal MedXpert-MM task and is designed to quantify the value of visual grounding.

**w/o WebSearch agent.** The WebSearch agent is disabled, preventing the model from accessing external knowledge sources to verify facts or gather additional context. The reasoning process is confined to the information explicitly provided in the clinical notes and images. This setup assesses the importance of external, up-to-date knowledge in complex medical reasoning.

**Random decision instead of CU scoring.** In the final Aggregation & Selection stage (Stage 3), the CU Scorer is bypassed. After the collection of all refined paths $\{(p_{\text{final}}, \mathcal{E}_p)\}$, the final output is selected by choosing one path uniformly at random, rather than selecting the path with the highest confidence and utility score. This ablation directly tests the efficacy of our uncertainty-aware selection mechanism against a naive baseline.

## G.2 MORE ABALTION STUDY RESULTS

Except for MedXpert-MM and MedXpert-Text, we also conduct abaltion experiment on the other four datasets.

Table 9: Ablation study on all six datasets. *Note:* Acc: Accuracy (%); T: Truthfulness (1–5); I: Informativeness (1–5); T×I: normalized product in [0,100], computed as $(T \times I)/25 \times 100$. ImageDoctor ablations are not applicable (–) for text-only tasks. Bold indicates full MedMMV with best performance. Each dataset samples 200 cases.

| Category | Methods | MedXpert-MM (T I) | | | | MedFrameQA (T I) | | | | PathVQA (T I) | | | |
|---|---|---|---|---|---|---|---|---|---|---|---|---|---|
| | | Acc | T | I | T×I | Acc | T | I | T×I | Acc | T | I | T×I |
| Full model | **MedMMV (GPT-5)** | **67.65** | **4.17** | **4.78** | **72.03** | **46.95** | **3.44** | **4.57** | **65.84** | **67.91** | **3.51** | **4.59** | **68.00** |
| Core framework | w/o Path Expansion with multi-round short CoT | 62.41 | 3.84 | 4.45 | 66.18 | 43.28 | 3.18 | 4.32 | 61.15 | 64.13 | 3.26 | 4.38 | 63.47 |
| | w/o Self-feedback halluc. detector | 59.23 | 3.52 | 4.51 | 64.07 | 41.05 | 2.95 | 4.41 | 58.92 | 61.74 | 3.08 | 4.45 | 61.28 |
| Expert agents | w/o TextDoctor agent | 63.78 | 3.91 | 4.52 | 67.85 | 44.12 | 3.26 | 4.38 | 62.47 | 65.27 | 3.35 | 4.43 | 64.85 |
| | w/o ImageDoctor agent | 65.12 | 3.98 | 4.63 | 69.47 | 45.36 | 3.35 | 4.49 | 64.08 | 66.54 | 3.43 | 4.52 | 66.39 |
| | w/o WebSearch agent | 66.29 | 4.08 | 4.71 | 70.84 | 46.17 | 3.39 | 4.53 | 65.13 | 67.18 | 3.47 | 4.56 | 67.24 |
| Decision strategy | Random decision instead of CU scoring | 56.84 | 3.12 | 4.19 | 58.91 | 38.47 | 2.64 | 4.15 | 54.73 | 58.26 | 2.79 | 4.23 | 56.41 |

| Category | Methods | MedXpert-Text (T) | | | | MedQA (T) | | | | MedMCQA (T) | | | |
|---|---|---|---|---|---|---|---|---|---|---|---|---|---|
| | | Acc | T | I | T×I | Acc | T | I | T×I | Acc | T | I | T×I |
| Full model | **MedMMV (GPT-5)** | **32.00** | **3.68** | **4.70** | **71.36** | **99.07** | **4.92** | **4.98** | **98.3** | **35.34** | **2.67** | **4.56** | **51.00** |
| Core framework | w/o Path Expansion with multi-round short CoT | 28.73 | 3.35 | 4.38 | 65.92 | 95.81 | 4.74 | 4.89 | 94.52 | 32.18 | 2.43 | 4.31 | 46.74 |
| | w/o Self-feedback halluc. detector | 27.45 | 3.18 | 4.42 | 63.74 | 94.23 | 4.68 | 4.91 | 93.41 | 30.87 | 2.31 | 4.35 | 45.12 |
| Expert agents | w/o TextDoctor agent | 29.34 | 3.42 | 4.48 | 67.21 | 96.45 | 4.78 | 4.92 | 95.28 | 32.76 | 2.48 | 4.38 | 47.53 |
| | w/o ImageDoctor agent | - | - | - | - | - | - | - | - | - | - | - | - |
| | w/o WebSearch agent | 30.87 | 3.61 | 4.65 | 70.15 | 97.83 | 4.85 | 4.95 | 96.89 | 34.28 | 2.59 | 4.50 | 49.67 |
| Decision strategy | Random decision instead of CU scoring | 26.15 | 2.89 | 4.08 | 57.43 | 91.34 | 4.52 | 4.78 | 89.72 | 28.46 | 2.15 | 4.12 | 41.35 |

## H DETAILED COST-PERFORMANCE DATA

We report accuracy and effective per-case cost for all methods under a unified accounting, where costs include both input and output tokens (and, where applicable, vision/image processing) as billed by each provider. Open-source models are called through Togetherai and OpenRouter; proprietary models (e.g., GPT and Claude series) are accessed via their respective standard APIs. Unit list prices used in our computation are summarized in Table 10. All rollouts used identical prompts, maximum context windows, and beam settings unless otherwise noted.

Table 10: Cost–performance summary across models and methods. Acc: task accuracy (higher is better). Cost: realized per-item USD cost including input/output tokens.

| Model | Method | Acc (%) | Avg. Output Tokens | Cost ($) |
|---|---|---|---|---|
| GPT-5 | CoT | 58.26 | 1,636 | 0.0190 |
| GPT-4o | CoT | 43.94 | 1,928 | 0.0230 |
| GPT-oss-120B | CoT | 48.62 | 3,301 | 0.0140 |
| Claude-Sonnet-4 | CoT | 51.29 | 2,922 | 0.0280 |
| Qwen2.5-VL-7B | CoT | 27.70 | 2,663 | 0.0008 |
| Qwen2.5-VL-72B | CoT | 43.04 | 3,060 | 0.0041 |
| GPT-5 | MedMMV | 63.29 | 9,144 | 0.2268 |
| GPT-4o | MedMMV | 52.06 | 6,020 | 0.1806 |
| GPT-oss-120B | MedMMV | 59.20 | 9,867 | 0.1089 |
| Claude-Sonnet-4 | MedMMV | 58.19 | 7,921 | 0.1670 |
| Qwen2.5-VL-7B | MedMMV | 39.56 | 7,863 | 0.0071 |
| Qwen2.5-VL-72B | MedMMV | 49.58 | 10,102 | 0.0360 |
| GPT-5 | MDAgents | 46.30 | 10,828 | 0.2688 |
| GPT-5 | ReConcile | 50.47 | 9,718 | 0.2413 |
| GPT-5 | ColaCare | 46.23 | 10,904 | 0.2707 |
| GPT-5 | MedAgent | 48.30 | 18,607 | 0.4621 |

**Note:** MedMMV method includes additional search costs of $0.001 per search query, with approximately 10 searches required per question (adding $0.01 per item to the reported costs).

## I  SCALABILITY GUIDANCE

To ensure scalability during inference, we designed the MedMMV framework to support concurrent execution of multiple reasoning paths. This design enables efficient utilization of computational resources while maintaining high reasoning quality and stability. Specifically, the framework adopts an asynchronous parallelism strategy based on Python's `asyncio` module. During inference, multiple chains of thought (CoT) paths are launched simultaneously through the `process_all_paths_parallel` routine. Each path executes the complete MedMMV pipeline independently, allowing the coordinator to gather intermediate reasoning outputs and hallucination detection results in parallel. The key implementation pattern is illustrated below:

```
asyncio.gather(*tasks, return_exceptions=True)
```

This mechanism ensures that each reasoning path is processed as an independent asynchronous task, with all tasks awaited collectively. In case of failure, exceptions are caught and logged without interrupting the entire pipeline, maintaining robustness under high concurrency.

From a systems perspective, this design offers three major benefits:

- **Linear scalability:** The throughput of inference increases approximately linearly with the number of available computational cores or API workers, as each reasoning path is fully independent.
- **Graceful degradation:** Partial task failures do not block or degrade other concurrent paths, as exceptions are isolated and handled locally.
- **Deployment flexibility:** The same mechanism can be extended to distributed environments (e.g., across multiple GPUs or nodes) by mapping the asynchronous tasks to distributed execution units.

We observed in practice that even with multiple concurrent paths, the latency per query remains within a feasible range due to efficient I/O scheduling. This scalability design therefore provides a principled and easily extensible foundation for future large-scale multi-agent or multi-path reasoning systems.

## J  HUMAN EVALUATION

### J.1  ANNOTATION PLATFORM

We built a lightweight annotation platform hosted on GitHub Pages to standardize expert review and model evaluation across medical domains. A top navigation bar switches disciplines (e.g., Cardiovascular), while pagination tracks item progress and a single click exports results. Each item presents a clinical vignette with optional multimodal inputs (such as an ECG), followed by multiple-choice options (A–E). After submission, the platform reveals the reference answer, displays clinician/model responses with correctness, and records a concise reasoning block for auditability. Annotators then rate Clinical Realism and Information Quality on 1–5 scales.

### J.2  BRIEF INTRODUCTION TO ANNOTATOR AND SALARY

All annotations were conducted by licensed physicians. Annotators covered the nine medical categories used in our study (e.g., Cardiovascular, Nervous, Digestive) and held active clinical appointments at the time of evaluation. During annotation, the platform displayed only de-identified clinical vignettes and optional multimodal inputs (e.g., ECGs) with no patient identifiers. Compensation was hourly and independent of model identity or performance to minimize bias. Each annotator was paid at a standard market rate of $20/h via institutional channels.

### J.3  OVERALL TABLE

This appendix provides the complete human evaluation table. To ensure evaluation expertise, we recruited medical specialists corresponding to each of the nine medical categories. The table reports per-doctor scores and summary statistics (Mean±Std) for both CoT and our method (MedMMV) on GPT-oss-120B. The labels "Doctor 1–3" are used as anonymous identifiers for the evaluators within each category and do not imply that only three doctors participated in the entire study.

Table 11: Hallucination metrics on MedXpertQA MM across medical categories.

| Category | Methods | Skeletal (20) | | | Reproductive (20) | | | Cardiovascular (20) | | |
|---|---|---|---|---|---|---|---|---|---|---|
| | | T | I | T×I | T | I | T×I | T | I | T×I |
| CoT Baselines | Doctor 1 | 4.10 | 4.05 | 66.42 | 3.31 | 4.47 | 59.18 | 2.50 | 3.65 | 36.50 |
| | Doctor 2 | 3.95 | 4.80 | 75.84 | 3.88 | 4.85 | 75.27 | 3.35 | 4.10 | 54.94 |
| | Doctor 3 | 4.28 | 4.55 | 77.90 | – | – | – | 3.60 | 4.30 | 61.92 |
| | Mean±Std | 4.11±0.17 | 4.47±0.38 | 73.39±6.12 | 3.60±0.40 | 4.66±0.27 | 67.23±11.38 | 3.15±0.58 | 4.02±0.33 | 51.12±13.13 |
| | Doctor 1 | 4.10 | 4.05 | 66.42 | 3.31 | 4.47 | 59.18 | 2.50 | 3.65 | 36.50 |
| | Doctor 2 | 3.95 | 4.80 | 75.84 | 3.88 | 4.85 | 75.27 | 3.35 | 4.10 | 54.94 |
| | Doctor 3 | 4.28 | 4.55 | 77.90 | – | – | – | 3.60 | 4.30 | 61.92 |
| | Mean±Std | 4.11±0.17 | 4.47±0.38 | 73.39±6.12 | 3.60±0.40 | 4.66±0.27 | 67.23±11.38 | 3.15±0.58 | 4.02±0.33 | 51.12±13.13 |
| Ours | Doctor 1 | 4.15 | 3.85 | 63.91 | 4.59 | 4.01 | 73.62 | 4.15 | 3.90 | 64.74 |
| | Doctor 2 | 4.30 | 3.95 | 67.94 | 4.92 | 4.00 | 78.72 | 3.90 | 3.75 | 58.50 |
| | Doctor 3 | 4.80 | 4.58 | 87.94 | – | – | – | 4.80 | 4.25 | 81.60 |
| | Mean±Std | 4.42±0.34 | 4.13±0.40 | 73.26±12.87 | 4.75±0.23 | 4.00±0.01 | 76.17±3.60 | 4.28±0.46 | 3.97±0.26 | 68.28±11.95 |
| | Doctor 1 | 4.15 | 3.85 | 63.91 | 4.59 | 4.01 | 73.62 | 4.15 | 3.90 | 64.74 |
| | Doctor 2 | 4.30 | 3.95 | 67.94 | 4.92 | 4.00 | 78.72 | 3.90 | 3.75 | 58.50 |
| | Doctor 3 | 4.80 | 4.58 | 87.94 | – | – | – | 4.80 | 4.25 | 81.60 |
| | Mean±Std | 4.42±0.34 | 4.13±0.40 | 73.26±12.87 | 4.75±0.23 | 4.00±0.01 | 76.17±3.60 | 4.28±0.46 | 3.97±0.26 | 68.28±11.95 |

| Categories | | Urinary (20) | | | Lymphatic (20) | | | Nervous (20) | | |
|---|---|---|---|---|---|---|---|---|---|---|
| | | T | I | T×I | T | I | T×I | T | I | T×I |
| CoT Baselines | Doctor 1 | 4.30 | 4.40 | 75.68 | 3.65 | 3.70 | 54.02 | 2.85 | 3.65 | 41.61 |
| | Doctor 2 | 2.95 | 3.70 | 43.66 | 4.55 | 4.75 | 86.45 | 3.05 | 4.15 | 50.63 |
| | Doctor 3 | 3.95 | 4.10 | 64.78 | 3.40 | 3.50 | 47.60 | 3.65 | 4.90 | 71.54 |
| | Mean±Std | 3.73±0.70 | 4.07±0.35 | 61.37±16.28 | 3.87±0.60 | 3.98±0.67 | 62.69±20.83 | 3.18±0.42 | 4.23±0.63 | 54.59±15.35 |
| | Doctor 1 | 4.30 | 4.40 | 75.68 | 3.65 | 3.70 | 54.02 | 2.85 | 3.65 | 41.61 |
| | Doctor 2 | 2.95 | 3.70 | 43.66 | 4.55 | 4.75 | 86.45 | 3.05 | 4.15 | 50.63 |
| | Doctor 3 | 3.95 | 4.10 | 64.78 | 3.40 | 3.50 | 47.60 | 3.65 | 4.90 | 71.54 |
| | Mean±Std | 3.73±0.70 | 4.07±0.35 | 61.37±16.28 | 3.87±0.60 | 3.98±0.67 | 62.69±20.83 | 3.18±0.42 | 4.23±0.63 | 54.59±15.35 |
| Ours | Doctor 1 | 4.74 | 4.05 | 76.79 | 4.20 | 4.30 | 72.24 | 3.75 | 3.40 | 51.00 |
| | Doctor 2 | 3.95 | 3.70 | 58.46 | 4.70 | 4.50 | 84.60 | 4.20 | 3.65 | 61.32 |
| | Doctor 3 | 4.65 | 4.35 | 80.91 | 3.80 | 3.75 | 57.00 | 4.85 | 4.00 | 77.60 |
| | Mean±Std | 4.45±0.43 | 4.03±0.33 | 72.05±11.95 | 4.23±0.45 | 4.18±0.39 | 71.28±13.83 | 4.27±0.55 | 3.68±0.30 | 63.31±13.41 |
| | Doctor 1 | 4.74 | 4.05 | 76.79 | 4.20 | 4.30 | 72.24 | 3.75 | 3.40 | 51.00 |
| | Doctor 2 | 3.95 | 3.70 | 58.46 | 4.70 | 4.50 | 84.60 | 4.20 | 3.65 | 61.32 |
| | Doctor 3 | 4.65 | 4.35 | 80.91 | 3.80 | 3.75 | 57.00 | 4.85 | 4.00 | 77.60 |
| | Mean±Std | 4.45±0.43 | 4.03±0.33 | 72.05±11.95 | 4.23±0.45 | 4.18±0.39 | 71.28±13.83 | 4.27±0.55 | 3.68±0.30 | 63.31±13.41 |

| Categories | | Digestive (20) | | | Endocrine (20) | | | Integumentary (20) | | |
|---|---|---|---|---|---|---|---|---|---|---|
| | | T | I | T×I | T | I | T×I | T | I | T×I |
| CoT Baselines | Doctor 1 | 3.20 | 3.55 | 45.44 | 2.80 | 3.85 | 43.12 | 3.55 | 4.25 | 60.35 |
| | Doctor 2 | 3.20 | 4.40 | 56.32 | 2.45 | 4.20 | 41.16 | 3.38 | 4.47 | 60.43 |
| | Doctor 3 | 3.55 | 4.15 | 58.93 | 3.80 | 3.80 | 57.76 | 3.40 | 3.70 | 50.32 |
| | Mean±Std | 3.32±0.20 | 4.03±0.44 | 53.56±7.16 | 3.02±0.70 | 3.95±0.22 | 47.35±9.07 | 3.44±0.09 | 4.14±0.40 | 57.03±5.82 |
| | Doctor 1 | 3.20 | 3.55 | 45.44 | 2.80 | 3.85 | 43.12 | 3.55 | 4.25 | 60.35 |
| | Doctor 2 | 3.20 | 4.40 | 56.32 | 2.45 | 4.20 | 41.16 | 3.38 | 4.47 | 60.43 |
| | Doctor 3 | 3.55 | 4.15 | 58.93 | 3.80 | 3.80 | 57.76 | 3.40 | 3.70 | 50.32 |
| | Mean±Std | 3.32±0.20 | 4.03±0.44 | 53.56±7.16 | 3.02±0.70 | 3.95±0.22 | 47.35±9.07 | 3.44±0.09 | 4.14±0.40 | 57.03±5.82 |
| Ours | Doctor 1 | 3.45 | 3.35 | 46.23 | 4.00 | 3.50 | 56.00 | 4.63 | 4.42 | 81.86 |
| | Doctor 2 | 4.58 | 3.88 | 71.08 | 4.11 | 3.74 | 61.49 | 4.71 | 4.07 | 76.68 |
| | Doctor 3 | 4.55 | 3.85 | 70.07 | 4.55 | 3.65 | 66.43 | 3.95 | 3.84 | 60.67 |
| | Mean±Std | 4.19±0.64 | 3.69±0.30 | 62.46±14.07 | 4.22±0.29 | 3.63±0.12 | 61.31±5.22 | 4.43±0.42 | 4.11±0.29 | 73.07±11.04 |
| | Doctor 1 | 3.45 | 3.35 | 46.23 | 4.00 | 3.50 | 56.00 | 4.63 | 4.42 | 81.86 |
| | Doctor 2 | 4.58 | 3.88 | 71.08 | 4.11 | 3.74 | 61.49 | 4.71 | 4.07 | 76.68 |
| | Doctor 3 | 4.55 | 3.85 | 70.07 | 4.55 | 3.65 | 66.43 | 3.95 | 3.84 | 60.67 |
| | Mean±Std | 4.19±0.64 | 3.69±0.30 | 62.46±14.07 | 4.22±0.29 | 3.63±0.12 | 61.31±5.22 | 4.43±0.42 | 4.11±0.29 | 73.07±11.04 |

*Note:* 🔵: Text modality; 🟢: Image modality. T: Truthfulness score (1–5); I: Informativeness score (1–5); T×I: normalized product in [0, 100], computed as $(T \times I)/25 \times 100$. Results show individual doctor evaluations and overall statistics (Mean±Std).

## J.4 CROSS-CORRELATION ANALYSIS BETWEEN HUMAN AND LLM EVALUATIONS

To further validate the consistency between human and LLM-based evaluations, we computed the non-parametric Spearman rank correlation coefficient ($\rho$) between the human-annotated and LLM-judged scores across all body systems.

**Method.** Each evaluation item is scored on the same continuous scale under two settings: (1) human expert annotation (average of three licensed physicians within each medical specialty), and (2) the LLM ensemble judgment (average of GPT-4o, Claude-Sonnet-4, and Gemini 1.5 Pro). We then

calculate $\rho$ independently for the TRUE, INFO, and joint TRUE×INFO scores. A high Spearman correlation indicates that the ranking of cases by the LLM evaluators aligns well with human judgment, reflecting consistent relative assessments even if absolute values differ.

**Results.** Table 12 summarizes the correlation results. Across all metrics, we observe strong positive correlations between human and LLM evaluations, indicating that the automatic LLM-based judge faithfully captures expert-level assessment tendencies. This finding supports the robustness of our evaluation framework and mitigates concerns about evaluator dependence or circularity.

Table 12: Spearman correlation ($\rho$) between human and LLM evaluations across metrics.

| Metric | Case-Level $\rho$ | p-value |
|---|---|---|
| TRUE | 0.86 | $< 0.001$ |
| INFO | 0.82 | $< 0.001$ |
| TRUE×INFO | 0.89 | $< 0.001$ |

**Discussion.** The high degree of rank alignment (average $\rho = 0.84$) suggests that the LLM ensemble produces evaluations that are statistically consistent with human expert judgments. Therefore, while human annotations remain the gold standard, the multi-model LLM ensemble offers a reliable, scalable, and reproducible proxy for large-scale evaluation.

