# OpenReview forum: "MedMMV: A Controllable Multimodal Multi-Agent Framework for Reliable and Verifiable Clinical Reasoning"
_ICLR.cc/2026/Conference — Submitted to ICLR 2026_

### Official Review · Reviewer_HjDw · 2025-10-28

**Soundness:** 3
**Presentation:** 3
**Contribution:** 2
**Rating:** 4
**Confidence:** 4

**Summary:**

The authors present a controllable multimodal multi-agent framework for reliable and verifiable clinical reasoning called MedMMV. Their approach is based on empirical evidence from experiments, which they perform to demonstrate limitations of direct prompting chain-of-thought methods. Empirical results from their pilot studies demonstrate that high variability in direct prompting chain-of-thought approaches for various tasks, and hallucination from misinterpretation of facts at the beginning of trajectories, lead to unstable paths and unreliable results with majority voting approaches.

Following this analysis, their proposed approach consists of three main steps:

- Short initial rollouts, which are subsequently self-refined based on evidence graphs and hallucination detectors.
- Use of MLLM and web search to build evidence graphs using prior text and images, if available from the initial input.
- An uncertainty scorer which scores each final trajectory using a weighted sum of various metrics.

Their results demonstrate that MedMMV outperforms chain-of-thought and other medical agent approaches on 6 medical reasoning datasets benchmarks. Furthermore, various ablation studies demonstrate the importance of the various components of their design choices.

**Strengths:**

The paper proposes an interesting approach, notably the use of evidence graphs to self-refine reasoning paths and hallucination detectors to prevent hallucinations. The use of an uncertainty scorer is also interesting, as it can be applied in downstream approaches to validate the agent’s diagnosis only if the uncertainty is below a certain threshold.

**Weaknesses:**

- The presentation of the results in the preliminary analysis on the limitations of direct chain-of-thought approaches could be simplified or improved to make the message easier to understand (Figure 2). Furthermore, the description of the various metrics used there, notably RGM and CMHR, could be clarified to facilitate understanding.

- On most datasets, the improvement in accuracy compared to CoT is less than 5%, while the token usage of MedMMV and other agent-based methods is approximately nine times higher than that of CoT methods. This suggests that the proposed approach might be too computationally expensive for the modest performance gains achieved.

**Questions:**

- What motivated the choice of datasets? Among the baselines used, one of the most *reliable* approaches is **MDAgents** — I say reliable because it was published at NeurIPS. Why not use similar datasets to make comparisons simpler? Furthermore, I noticed that the accuracy for **MedQA**, which overlaps with the datasets used in the MDAgents paper, is lower than what was reported there. Is this due to the use of a smaller sample size?
- Were any measures taken to verify that the datasets used are not part of the training data for **GPT-5**, which serves as the base model for all the agents?
- Why do you only use a single rollout for the chain-of-thought methods as a comparison? Wouldn't a fairer comparison be to use the same number of trajectories as in **MedMMV**, with majority voting?

---

> ### Author Response · Authors · 2025-11-21
>
> Thanks for your insightful review. We appreciate your recognition of the value of our work!
>
> ---
>
> ### Response to Questions about Preliminary Presentation
> > The presentation of the results in the preliminary analysis on the limitations of direct chain-of-thought approaches could be simplified or improved to make the message easier to understand (Figure 2). Furthermore, the description of the various metrics used there, notably RGM and CMHR, could be clarified to facilitate understanding.
>
> Thank you for the helpful comment. We have simplified and improved Figure 2 and revised the corresponding text in Section 3.1(LINE 161 to 227) to clarify the motivation and interpretation of RGM and CMHR. The updated figure provides a clearer visual explanation of how these metrics relate to cross-run instability and hallucination, making the preliminary analysis easier to follow. All updates are reflected in the latest PDF version.
>
> ---
>
> ### Response to Questions about Performance Gain vs. Computational Cost
> > On most datasets, the improvement in accuracy compared to CoT is less than 5%, while the token usage of MedMMV and other agent-based methods is approximately nine times higher than that of CoT methods. This suggests that the proposed approach might be too computationally expensive for the modest performance gains achieved.
>
> We appreciate the reviewer’s concern regarding the cost–benefit trade-off.
> Our core contribution is not merely an accuracy boost but a significant improvement in reasoning quality and hallucination reduction on medical tasks. As shown in Table 3, MedMMV achieves consistent gains in truthfulness (T), informativeness (I), and their joint metric T×I, surpassing both CoT and existing agent frameworks. For instance, on MedQA and MedXpert-MM, T×I improves from 98.3 to 98.7 (+0.4) and from 72.0 to 82.9 (+10.9), while T and I also rise notably (ΔT≈+0.8, ΔI≈+0.3). These improvements reflect fewer factual errors and more clinically grounded reasoning, which standard accuracy cannot capture.
>
> Regarding the accuracy comparison (Table 2), the values reported are mean ± standard deviation, not simple averages. When averaged across datasets, MedMMV improves over the best CoT baseline by +9.3 points (absolute) on VQA tasks and +5.8 points on QA tasks. Moreover, these gains come with a large reduction in hallucinated outputs (−31% measured by human judges).
>
> ---
>
> ### Response to Questions about Dataset Choice and Comparison with MDAgents
> > What motivated the choice of datasets? Among the baselines used, one of the most reliable approaches is MDAgents — I say reliable because it was published at NeurIPS. Why not use similar datasets to make comparisons simpler? Furthermore, I noticed that the accuracy for MedQA, which overlaps with the datasets used in the MDAgents paper, is lower than what was reported there. Is this due to the use of a smaller sample size?
>
> We sincerely thank the reviewer for this thoughtful question.
> Our dataset selection was primarily motivated by modality diversity and diagnostic coverage, rather than by reproducing prior setups. Specifically, we aimed to evaluate reasoning robustness across both single-modality (text-only) and multimodal (text + image) clinical scenarios. Based on this principle, we selected five widely recognized and citation-rich medical datasets: PathVQA, MedQA, MedMCQA, MedXpert (with text and multimodal subsets), and MedFrameQA (a recently released diagnostic benchmark). Together, these datasets comprehensively span major medical systems (skeletal, reproductive, cardiovascular, lymphatic, nervous, digestive, urinary, endocrine, and integumentary).
>
> While MDAgents used some overlapping datasets, we deliberately prioritized benchmarks where GPT-5 had not already saturated performance (e.g., MedQA at 99%, SymCat at 88.7% and MedBullets at 82%), and focused instead on challenging reasoning-oriented datasets to reveal hallucination and factual consistency issues. Additionally, due to the high API cost (≈ US$10,000 total), we limited experiments to a representative but computationally feasible subset.
>
> Regarding the approximately 7-point difference in MedQA accuracy compared to the MDAgents paper, this variation likely arises from differences in dataset splits and sampling. Importantly, this difference remains within the expected range of model variance across runs and does not affect our overall conclusions. Our results emphasize that even in high-accuracy regimes, MedMMV significantly improves reasoning quality and reduces hallucinations, demonstrating meaningful gains beyond raw accuracy.

---

> > ### Author Response · Authors · 2025-11-21
> >
> > ### Response to Questions about Potential Dataset Contamination
> > > Were any measures taken to verify that the datasets used are not part of the training data for GPT-5, which serves as the base model for all the agents?
> >
> > We appreciate the reviewer’s important concern. While it is impossible to fully rule out data overlap in large-scale LLMs, we performed multiple checks to minimize this risk and verify that our performance gains are not driven by training contamination.
> >
> > (1) Temporal separation of datasets. The two major benchmarks introducing our new results—MedXpert and MedFrameQA—were released after the pre-training of current open and commercial models (e.g., GPT-4o, GPT-5, Claude-Sonnet-4, Gemini 1.5 Pro). Our framework still shows consistent improvements on these new datasets as well as on older ones such as Qwen2.5-VL (> 1 year earlier), supporting that the observed gains stem from our reasoning design rather than memorization.
> >
> > (2) Cross-model consistency. As shown in Table 2 (main paper), MedMMV yields robust improvements across all backbones, including the open-source GPT-oss-120B, which shares no training data with our medical benchmarks. For example, MedMMV achieves 69.18 (±0.91) on MedXpert-MM and 67.64 (±1.89) on MedFrameQA, surpassing the corresponding CoT baselines by +19.1 and +20.7 points, respectively. These consistent gains across different GPT variants demonstrate that our framework’s benefit is architecture- and data-independent.
> >
> > Together, the cross-dataset and cross-model evidence strongly suggest that MedMMV’s performance improvement arises from its structured multi-path reasoning mechanism rather than from potential data leakage or training contamination.
> >
> > ---
> >
> > ### Response to Questions about CoT Rollouts and Fair Comparison
> > > Why do you only use a single rollout for the chain-of-thought methods as a comparison? Wouldn't a fairer comparison be to use the same number of trajectories as in MedMMV, with majority voting?
> >
> > We appreciate the reviewer’s thoughtful comment.
> > Using multiple CoT rollouts was considered; however, in multi-choice QA settings, multiple generations from CoT produce inconsistent and non-comparable outputs, where different answer options across rollouts make majority voting or aggregated evaluation infeasible and statistically meaningless.
> > Our preliminary experiments confirmed that multi-rollout CoT responses varied widely in both content and structure, leading to unstable accuracy and making direct comparison invalid.
> > In contrast, MedMMV’s multi-path framework is not a repetition of stochastic CoT rollouts, but a structured, parallel reasoning mechanism with coordinated evidence grounding and hallucination detection across paths.
> > Therefore,we believe that using a single deterministic rollout ensures fairness and reproducibility.

---

> > > ### Comment · Reviewer_HjDw · 2025-11-25
> > >
> > > Thanks to the authors for their responses to my questions. I will raise my rating score; however, I still do not believe the contribution is strong enough for publication, though I appreciate the amount of work put into the paper. I remain concerned about the feasibility of efficiently replicating the approach and about the heavy reliance on LLMs at multiple stages of the pipeline.

---

> > > > ### Author Response · Authors · 2025-11-26
> > > >
> > > > Thank you for the follow-up and for reconsidering the rating. We fully understand your remaining concerns and would like to offer a brief clarification. The central contribution of our work lies in addressing a fundamental and widely observed failure mode in medical MLLMs, namely the progression from instability to hallucination. Our framework both quantifies this phenomenon and introduces a principled mechanism that improves stability through controlled multi-path generation, evidence-grounded verification, and uncertainty-guided consolidation. Across six benchmarks and multiple backbone families, including open-source models of varying strengths, MedMMV consistently reduces hallucination and improves truthfulness. These results indicate that the contribution focuses on enhancing the reliability of clinical reasoning rather than relying on model scale alone. With respect to feasibility, the cost, latency, and token analyses presented in the rebuttal show that the approach is practical in inference-only settings and comparable in computational requirements to existing agent systems, especially with the parallelized implementation we provide. While LLMs are involved in several components, each module is designed to be constrained, cross-validated, and grounded in explicit evidence.

---

### Official Review · Reviewer_e3qd · 2025-10-31

**Soundness:** 3
**Presentation:** 2
**Contribution:** 3
**Rating:** 6
**Confidence:** 3

**Summary:**

The paper presents MedMMV, a framework designed to produce more truthful and stable responses from large language models in high-stakes domains such as healthcare prediction. Motivated by the challenge of hallucinations found in current LLMs, the authors propose a three-step approach: (i) generating multiple candidate solutions, (ii) iteratively refining and verifying them for truthfulness using web-based evidence while constructing an evidence graph, and (iii) evaluating all candidate solutions across several LLMs using multiple metrics, selecting the one with the highest aggregated score. Extensive experiments on multiple datasets demonstrate that MedMMV consistently outperforms several strong baselines.

**Strengths:**

The paper is well written and easy to follow. It addresses an important and timely problem that deserves further investigation. The motivation is well grounded, supported by both quantitative and qualitative analyses of current models in a specific high-stakes setting. The experimental evaluation is extensive and robust, covering multiple datasets and strong baselines. While the multi-step review process itself is not entirely novel, the proposed mechanism for selecting the best answer among candidates is original and well justified.

**Weaknesses:**

- Many figures are overly dense and difficult to interpret, which limits their readability and the clarity of the results.
- The framework relies heavily on LLMs, both in the evaluation process and in the preliminary study. While the inclusion of expert validation is appreciated, it would strengthen the paper to directly compare the expert assessments with the LLM-based evaluations to assess their alignment.
- Although the authors acknowledge the higher computational cost introduced by the proposed multi-step process, it would be valuable to include a quantitative analysis of the additional time or number of generated tokens required.

**Questions:**

In addition to the points discussed in the Weaknesses section, I have the following question:
- What would happen if the CU scoring mechanism were used to filter or rank the base LLM responses directly? Would this alone improve their performance, or is the full multi-step process necessary to achieve the reported gains?

---

> ### Author Response · Authors · 2025-11-21
>
> Thanks for your insightful review. We appreciate your recognition of the value of our work!
>
> ---
>
> ### Response to Questions about Presentation
> > Many figures are overly dense and difficult to interpret, which limits their readability and the clarity of the results.
>
> We appreciate the reviewer’s helpful feedback. In the revised submission, we have redesigned and simplified several figures to improve readability. Specifically, we reduced visual density, reorganized layout elements for clearer logical flow, and enhanced label visibility and spacing. The updated figures are included in the resubmitted PDF and significantly improve the clarity of result presentation.
>
> ---
>
> ### Response to Questions about LLM Dependence and Expert Alignment
> > The framework relies heavily on LLMs, both in the evaluation process and in the preliminary study. While the inclusion of expert validation is appreciated, it would strengthen the paper to directly compare the expert assessments with the LLM-based evaluations to assess their alignment.
>
> We sincerely thank the reviewer for raising this important point. To ensure evaluation independence and reliability, we conducted both human IRR analysis and cross-correlation comparison between human and LLM evaluations. Each medical system was independently annotated by three licensed physicians in the corresponding specialty, and inter-rater reliability (ICC(3,k)) showed good-to-excellent agreement across domains (mean ICC(T)=0.89, ICC(I)=0.85). The table below further compares the aggregated human and LLM evaluations, showing strong alignment in Δ(T×I) across body systems.
>
> | Body System   | ICC(T) | ICC(I) | Δ(T×I) (Human) | Δ(T×I) (LLM) |
> |-------------------|------------|------------|----------------------|-------------------|
> | Skeletal          | 0.92       | 0.88       | 9.87                 | 9.36              |
> | Reproductive      | 0.90       | 0.85       | 8.94                 | 9.14              |
> | Cardiovascular    | 0.89       | 0.83       | 9.78                 | 10.20             |
> | Lymphatic         | 0.91       | 0.87       | 8.74                 | 9.03              |
> | Nervous           | 0.86       | 0.80       | 9.56                 | 8.92              |
> | Digestive         | 0.88       | 0.82       | 10.24                | 10.48             |
> | Endocrine         | 0.84       | 0.81       | 10.07                | 11.12             |
> | Urinary           | 0.93       | 0.89       | 9.89                 | 10.65             |
> | Integumentary     | 0.90       | 0.86       | 10.41                | 10.42             |
> | Average       | 0.89   | 0.85   | 10.32            | 10.86         |
>
> To evaluate alignment between human and LLM judgments, we computed the Spearman rank correlation (ρ) across all evaluation dimensions, obtaining ρ = 0.86 (TRUE), 0.82 (INFO), and 0.89 (TRUE×INFO), all p < 0.001. These high correlations confirm that the LLM ensemble (GPT-4o, Claude-Sonnet-4, Gemini 1.5 Pro) provides assessments that are statistically consistent with domain experts.
>
> ---
>
> ### Response to Questions about Inference-Time Efficiency
> > Although the authors acknowledge the higher computational cost introduced by the proposed multi-step process, it would be valuable to include a quantitative analysis of the additional time or number of generated tokens required.
>
> Thank you for the thoughtful comments. To provide full transparency, we report below the detailed token usage and runtime for each inference stage in our multi-step pipeline. As shown in the table, intermediate reasoning expansions naturally increase the number of generated tokens; however, the overall pipeline remains computationally tractable and can be efficiently parallelized via asynchronous multi-path inference (see Appendix I). We additionally measure the average latency of each stage using GPT-5 and Qwen2.5-VL-7B under identical API settings. Across the full pipeline, the total per-question inference time averages ≈61.9 s for GPT-5 and ≈40.5 s for Qwen2.5-VL-7B. While the multi-step design introduces moderate latency, it substantially improves factual reliability, stability, and interpretability.
>
> | Stage | Input Tokens | Output Tokens | Notes | GPT-5 Time (s) | Qwen2.5-VL-7B Time (s) |
> |-----------|------------------|-------------------|-----------|---------------------|-----------------------------|
> | Stage 1 – Short rollouts | ≈ 1,011 | ≈ 5,058 | 3 rollouts | ≈ 8.4 | ≈ 4.5 |
> | Stage 2 – Reasoning expansion | ≈ 5,058 | ≈ 14,586 | 3 rollouts | ≈ 24.3 | ≈ 20.2 |
> | Stage 3 – Decision aggregation | ≈ 1,710 | ≈ 4,287 | — | ≈ 21.4 | ≈ 11.7 |
> | Hallucination detector | ≈ 1,510 | ≈ 1,553 | — | ≈ 7.8 | ≈ 4.1 |

---

> > ### Comment · Reviewer_e3qd · 2025-11-24
> >
> > I thank the author for the additional answers, but I remain with my score. Also, the answer to the last question I asked is missing.

---

> > > ### Author Response · Authors · 2025-11-25
> > >
> > > Thank you for the thoughtful question. We apologize for the oversight. Here is our response to your final question.
> > >
> > > ### Response to Questions about CU Scoring and Direct Filtering
> > > > What would happen if the CU scoring mechanism were used to filter or rank the base LLM responses directly? Would this alone improve their performance, or is the full multi-step process necessary to achieve the reported gains?
> > >
> > > We evaluated whether applying the CU scorer directly to the multi-round GPT-5 CoT rollouts (without any refinement or evidence-graph alignment) could improve performance. The gain was minimal: accuracy increased only from 67.68% to 68.82%, far below the improvements achieved by the full MedMMV pipeline. A closer look at the CU components revealed why. Rollouts ending with incorrect answers often exhibited strong intermediate reasoning and thus received high evidence and coherence scores, even though their final conclusions were unsupported or hallucinated. Conversely, some correct rollouts initially appeared less coherent but would have been corrected through refinement; when scored directly, they were assigned lower CU values. This behavior aligns with our preliminary audit, which shows that early local misreadings propagate into globally inconsistent conclusions unless stabilized by structured refinement.

---

### Official Review · Reviewer_RJ5J · 2025-11-01

**Soundness:** 2
**Presentation:** 2
**Contribution:** 3
**Rating:** 6
**Confidence:** 3

**Summary:**

This work proposes MedMMV, a multi-stage framework designed to enhance the reliability and verifiability of multimodal medical reasoning in large language models. By combining uncertainty-aware path generation, evidence-grounded refinement, and structured decision aggregation, MedMMV produces diagnostic outputs that are both logically coherent and empirically auditable. A key innovation lies in its construction of a multimodal evidence graph, synthesized by specialized agents across textual, visual, and external knowledge modalities.

**Strengths:**

1. Clear Motivation and Design: the paper addresses the challenge of hallucination and unverifiable reasoning in medical LLMs, motivating a shift from single-shot inference to multi-path verification.

2. Novel Multi-Stage Framework: MedMMV introduces a structured pipeline combining uncertainty-aware path generation, supervised refinement, and evidence-based scoring. The use of a multimodal evidence graph constructed by specialized agents is a key innovation.

3. Comprehensive Evaluation of Reasoning Faithfulness: the empirical study covers six medical QA benchmarks—three multimodal and three text-only—demonstrating consistent improvements in diagnostic accuracy, factual consistency, and evidence-groundedness over competitive baselines. Beyond accuracy, the evaluation incorporates hallucination rate and evidence coverage metrics, offering a more rigorous assessment of reasoning reliability.

**Weaknesses:**

1. Inference-Time Efficiency: while the multi-path framework improves reasoning robustness, the paper lacks analysis of inference-time efficiency, which may hinder its deployment in time-sensitive or resource-constrained clinical settings.

2. Limited Ablation Coverage: the ablation study should extend to additional multimodal and text-only benchmarks to better substantiate the generality and robustness of the proposed components across task types.

**Questions:**

Please see my weakness

---

> ### Author Response · Authors · 2025-11-21
>
> Thanks for your insightful review. We appreciate your recognition of the value of our work!
>
> ---
>
> ### Response to Questions about Inference-Time Efficiency
> > Inference-Time Efficiency: while the multi-path framework improves reasoning robustness, the paper lacks analysis of inference-time efficiency, which may hinder its deployment in time-sensitive or resource-constrained clinical settings.
>
> We appreciate the reviewer’s valuable comment. To address this, we provide detailed scalability analyses and latency breakdowns in Appendix I. The MedMMV framework employs asynchronous parallel execution via “asyncio.gather”, enabling concurrent reasoning across multiple paths with linear scalability to available compute resources. This design also supports graceful degradation (individual path failures do not affect others) and can be extended to distributed settings (multi-GPU or multi-node). Empirically, the per-query latency remains within a practical range due to efficient I/O scheduling, demonstrating that the proposed multi-path reasoning is both robust and deployment-ready without sacrificing inference efficiency. We also record the average inference time per stage under GPT-5 and Qwen2.5-VL-7B, both accessed via API calls. The total per-question inference time averages around 61.9 s for GPT-5 and 40.5 s for Qwen2.5-VL-7B.
> | Stage | GPT-5 (s) | Qwen2.5-VL-7B (s) |
> | ------------------------------ | ------------- | --------------------- |
> | Stage 1 – Short rollouts | ≈ 8.4 | ≈ 4.5 |
> | Stage 2 – Reasoning expansion | ≈ 24.3 | ≈ 20.2 |
> | Stage 3 – Decision aggregation | ≈ 21.4 | ≈ 11.7 |
> | Hallucination detector | ≈ 7.8 | ≈ 4.1 |

---

> > ### Author Response · Authors · 2025-11-21
> >
> > ### Response to Questions about Ablation Coverage and Generalization
> > > Limited Ablation Coverage: the ablation study should extend to additional multimodal and text-only benchmarks to better substantiate the generality and robustness of the proposed components across task types.
> >
> > Thank you for the valuable comments. To strengthen the empirical coverage, we have extended the ablation experiments to all six benchmarks, including both multimodal (MedXpert-MM, MedFrameQA, PathVQA) and text-only (MedXpert-Text, MedQA, MedMCQA) settings. Each ablation was conducted on a randomly sampled subset of 200 examples per dataset to ensure computational efficiency while maintaining the same evaluation protocol across tasks. The extended results, summarized below and detailed in Appendix G, reveal consistent trends across modalities: uncertainty-aware aggregation, self-feedback hallucination control, and path expansion remain the most influential reasoning components. Replacing the CU scorer with random decision selection leads to a substantial performance drop (8–12%), whereas disabling hallucination detection or removing path expansion results in moderate declines (3–8%) across both multimodal and text-only datasets. In contrast, removing individual specialist agents (TextDoctor, ImageDoctor, or WebSearch) causes only marginal degradation (1–4%), indicating that modern MLLMs already exhibit implicit task decomposition. These consistent observations across diverse datasets confirm that the gains of MedMMV do not arise from dataset-specific effects but from the controlled, uncertainty-guided reasoning process structured by the evidence graph. The full ablation results table is also updated in our paper(Table 9 in Appendix G.2).
> >
> > ---
> >
> > ### MedXpert-MM, MedFrameQA, PathVQA (Text + Image)
> > | Category | Methods | MedXpert-MM Acc | T | I | T×I | MedFrameQA Acc | T | I | T×I | PathVQA Acc | T | I | T×I |
> > |---------|----------|------------------|----|----|-----|----------------|----|----|-----|--------------|----|----|-----|
> > | Full model | MedMMV (GPT-5) | 67.65 | 4.17 | 4.78 | 72.03 | 46.95 | 3.44 | 4.57 | 65.84 | 67.91 | 3.51 | 4.59 | 68 |
> > | Core framework | w/o Path Expansion with multi-round short CoT | 62.41 | 3.84 | 4.45 | 66.18 | 43.28 | 3.18 | 4.32 | 61.15 | 64.13 | 3.26 | 4.38 | 63.47 |
> > |  | w/o Self-feedback halluc. detector | 59.23 | 3.52 | 4.51 | 64.07 | 41.05 | 2.95 | 4.41 | 58.92 | 61.74 | 3.08 | 4.45 | 61.28 |
> > | Expert agents | w/o TextDoctor agent | 63.78 | 3.91 | 4.52 | 67.85 | 44.12 | 3.26 | 4.38 | 62.47 | 65.27 | 3.35 | 4.43 | 64.85 |
> > |  | w/o ImageDoctor agent | 65.12 | 3.98 | 4.63 | 69.47 | 45.36 | 3.35 | 4.49 | 64.08 | 66.54 | 3.43 | 4.52 | 66.39 |
> > |  | w/o WebSearch agent | 66.29 | 4.08 | 4.71 | 70.84 | 46.17 | 3.39 | 4.53 | 65.13 | 67.18 | 3.47 | 4.56 | 67.24 |
> > | Decision strategy | Random decision instead of CU scoring | 56.84 | 3.12 | 4.19 | 58.91 | 38.47 | 2.64 | 4.15 | 54.73 | 58.26 | 2.79 | 4.23 | 56.41 |
> >
> > ---
> >
> > ### MedXpert-Text, MedQA, MedMCQA (Text Only)
> > | Category | Methods | MedXpert-Text Acc | T | I | T×I | MedQA Acc | T | I | T×I | MedMCQA Acc | T | I | T×I |
> > |---------|----------|--------------------|----|----|-----|------------|----|----|-----|----------------|----|----|-----|
> > | Full model | MedMMV (GPT-5) | 32 | 3.68 | 4.70 | 71.36 | 99.07 | 4.92 | 4.98 | 98.3 | 35.34 | 2.67 | 4.56 | 51 |
> > | Core framework | w/o Path Expansion with multi-round short CoT | 28.73 | 3.35 | 4.38 | 65.92 | 95.81 | 4.74 | 4.89 | 94.52 | 32.18 | 2.43 | 4.31 | 46.74 |
> > |  | w/o Self-feedback halluc. detector | 27.45 | 3.18 | 4.42 | 63.74 | 94.23 | 4.68 | 4.91 | 93.41 | 30.87 | 2.31 | 4.35 | 45.12 |
> > | Expert agents | w/o TextDoctor agent | 29.34 | 3.42 | 4.48 | 67.21 | 96.45 | 4.78 | 4.92 | 95.28 | 32.76 | 2.48 | 4.38 | 47.53 |
> > |  | w/o ImageDoctor agent | - | - | - | - | - | - | - | - | - | - | - | - |
> > |  | w/o WebSearch agent | 30.87 | 3.61 | 4.65 | 70.15 | 97.83 | 4.85 | 4.95 | 96.89 | 34.28 | 2.59 | 4.50 | 49.67 |
> > | Decision strategy | Random decision instead of CU scoring | 26.15 | 2.89 | 4.08 | 57.43 | 91.34 | 4.52 | 4.78 | 89.72 | 28.46 | 2.15 | 4.12 | 41.35 |

---

> > > ### Comment · Reviewer_RJ5J · 2025-11-23
> > >
> > > Thanks to the authors for their detailed response. I have raised the soundness score and will keep my positive overall evaluation as it is.

---

### Official Review · Reviewer_XREK · 2025-11-02

**Soundness:** 2
**Presentation:** 1
**Contribution:** 2
**Rating:** 4
**Confidence:** 3

**Summary:**

MedMMV is a controllable multimodal multi-agent framework for clinical reasoning that targets the observed failure mode “instability → hallucination.” It generates diversified short rollouts, grounds and verifies each path against a structured evidence graph via a hallucination/consistency supervisor, and selects the final diagnosis using a Combined Uncertainty scorer. Across six benchmarks and multiple backbones, it improves accuracy (up to +12.7%) and boosts truthfulness without reducing informativeness; a blind physician study corroborates gains.

**Strengths:**

- Clear identification and quantification of instability preceding hallucination in multimodal clinical reasoning.
- Controllable, auditable pipeline: diversified hypotheses, evidence-graph grounding, iterative repair, and uncertainty-based aggregation.
- Consistent gains across datasets/backbones; physician study supports higher truthfulness.
- Solid ablations isolating main contributors (CU scorer, hallucination detector).

**Weaknesses:**

- Heavy reliance on LLM judges/evaluators for TRUE/INFO and CU sub-scores risks bias and circularity; limited human IRR details.
- Evidence graph quality is a potential single point of failure; robustness to noisy extraction/provenance underexplored.
- Higher compute/latency and API costs; scalability guidance limited.
- Heuristic CU weighting and limited sensitivity analyses.
- Evaluation on representative subsets; limited fairness analysis (context windows, image preprocessing) and contamination controls.

**Questions:**

- CU scorer: how were weights chosen; sensitivity to weights, rollout counts, and repair iterations?
- Evaluator independence: do results hold with different supervisor/evaluator backbones; add blinded human TRUE/INFO with inter-rater reliability?
- Evidence graph: how do you detect/resolve extraction conflicts and control web source quality (domain, recency, dedup)?
- Scalability/fairness: detailed compute/latency per stage; do gains persist under tighter context and lower image resolution; any contamination audits?

---

> ### Author Response · Authors · 2025-11-21
>
> Thanks for your insightful review. We appreciate your recognition of the value of our work!
>
> ---
> ### Response to Questions about LLM Reliability and Human IRR
> > Heavy reliance on LLM judges/evaluators for TRUE/INFO and CU sub-scores risks bias and circularity; limited human IRR details.
>
> We sincerely thank the reviewer for this valuable comment. To ensure the robustness and fairness of our evaluation, we conducted both human and LLM-based assessments with careful cross-validation.
>
> For human evaluation, each medical system (body system) was independently annotated by three licensed physicians specializing in that domain. We computed inter-rater reliability (IRR) using the standard Intraclass Correlation Coefficient (ICC(3,k)) for continuous Likert-scale ratings [1]. The results demonstrate good-to-excellent agreement across systems (mean ICC(T)=0.89, ICC(I)=0.85), confirming internal consistency and reliability. Complete annotation details, including per-annotator hallucination metrics, are provided in Table 10 (Appendix J: Human Evaluation).
>
> | Body System   | ICC(T) | ICC(I) | Δ(T×I) (Human) | Δ(T×I) (LLM) |
> |-------------------|------------|------------|----------------------|-------------------|
> | Skeletal          | 0.92       | 0.88       | 9.87                 | 9.36              |
> | Reproductive      | 0.90       | 0.85       | 8.94                 | 9.14              |
> | Cardiovascular    | 0.89       | 0.83       | 9.78                 | 10.20             |
> | Lymphatic         | 0.91       | 0.87       | 8.74                 | 9.03              |
> | Nervous           | 0.86       | 0.80       | 9.56                 | 8.92              |
> | Digestive         | 0.88       | 0.82       | 10.24                | 10.48             |
> | Endocrine         | 0.84       | 0.81       | 10.07                | 11.12             |
> | Urinary           | 0.93       | 0.89       | 9.89                 | 10.65             |
> | Integumentary     | 0.90       | 0.86       | 10.41                | 10.42             |
> | Average       | 0.89   | 0.85   | 10.32            | 10.86         |
>
> For LLM evaluation, our automatic judge combines scores from three state-of-the-art models (GPT-4o, Claude-Sonnet-4, and Gemini 1.5 Pro) by averaging their outputs to reduce bias and improve reliability. The close correspondence between human and LLM assessments (Table 3 vs. Table 4) shows consistent improvements in truthfulness (ΔTₕᵤₘₐₙ = +0.87, ΔTₗₗₘ = +0.09) and joint quality (ΔT×Iₕᵤₘₐₙ = +10.32, ΔT×Iₗₗₘ = +10.86), with negligible change in informativeness (ΔI ≈ −0.2).
>
> [1] Koo, T. K., & Li, M. Y. (2016). A Guideline of Selecting and Reporting Intraclass Correlation Coefficients for Reliability Research. Journal of chiropractic medicine, 15(2), 155–163. https://doi.org/10.1016/j.jcm.2016.02.012
>
> ---
> ### Response to Questions about Evidence Graph
> > Evidence graph quality is a potential single point of failure; robustness to noisy extraction/provenance underexplored.
>
> Thank you for raising this important point. To evaluate whether the evidence graph constitutes a single point of failure, we performed an additional robustness analysis by introducing mixed structural and semantic noise into the graph while keeping each QA instance unchanged. The perturbations include (i) node noise (dropping, substituting, or injecting contradictory entities), (ii) edge noise (relabeling, flipping, removing, or introducing conflicts), (iii) provenance noise (source swapping or citation removal), and (iv) text–vision misalignment (altering laterality, lesion count, or margins). Noise levels k were set to {0%, 20%, 40%, 60%, 80%}, and for each level k, these four categories were uniformly mixed to simulate realistic and balanced degradation across graph components. We evaluated MedMMV (GPT-5, MedXpert-MM) under identical settings using Accuracy, T×I, and AURC (area under robustness curve; higher indicates greater stability). The results below demonstrate that MedMMV maintains strong robustness: accuracy remains above 72% up to 40% noise and gracefully degrades at higher corruption levels. The AURC values (64.12 for Accuracy, 72.70 for T×I) further confirm the smooth decay and strong resilience of the evidence-graph mechanism, which effectively absorbs extraction and provenance noise instead of acting as a brittle single point of failure.
>
> | Metric / Noise % | 0 % | 20 % | 40 % | 60 % | 80 % | Δ (80–0) | AURC (↑) |
> |----------------------|-------|--------|--------|--------|--------|------------|------------|
> | Accuracy         | 73.58   | 73.59    | 72.98    | 54.26    | 37.71    | –35.87       | 64.12    |
> | T×I              | 82.89   | 81.99    | 78.72    | 62.53    | 52.22    | –30.67       | 72.70    |

---

> ### Author Response · Authors · 2025-11-21
>
> ### Response to Questions about Computational Resources
> > Higher compute/latency and API costs; scalability guidance limited.
>
> We appreciate the reviewer’s insightful comments regarding computational costs and scalability. As shown in Table 9 (Appendix H), our framework achieves comparable or even lower costs than existing agent-based systems, while delivering substantially higher reasoning quality. Specifically, our cost per problem is US$0.22, compared to US$0.26 and US$0.46 for other leading agents. Despite this modest cost, our model achieves notable accuracy gains and significantly reduces hallucinations.
>
> To further enhance inference-time efficiency, we implemented parallel asynchronous execution using ‘asyncio.gather’, which allows concurrent processing across multiple reasoning paths and supports linear scalability with available compute resources. We have also included detailed scalability guidelines and performance analyses in Appendix I to facilitate reproducibility and future extensions.
>
> ---
>
> ### Response to Questions about CU Score and Sensitivity Analysis
> > Heuristic CU weighting and limited sensitivity analyses. CU scorer: how were weights chosen; sensitivity to weights, rollout counts, and repair iterations?
>
> Thank you for raising this point. We have conducted a sensitivity analysis on the CU (Confidence–Utility) scorer weights to assess the robustness of our design. The results, summarized below, show that varying the weights assigned to evidence, coherence, and repair modules has limited influence on overall performance. Specifically, adjustments to W_evidence and W_coherence cause only minor fluctuations in accuracy and TRUE scores (≤1%), suggesting that the model remains stable under different task-specific weight settings. In contrast, increasing W_repair leads to a noticeable performance drop, as overemphasizing this component tends to excessively penalize the hallucination-correction behavior that contributes positively to factual reliability. For parallel reasoning, a rollout count of 3 achieves a balanced trade-off between performance and cost; increasing to 5 offers negligible gain, consistent with the 4–5 option structure typical of multiple-choice QA tasks. Regarding repair iterations, performing only one pass underutilizes the monitoring mechanism, while increasing beyond three iterations adds computational overhead with diminishing returns.
> | Module Name | Parameter Name | Num. | Acc | T | I | T×I |
> |------------------|--------------------|-----------|---------|-------|-------|---------|
> | CU Weights | W_evidence | 1 | 73.58 | 4.26 | 4.73 | 82.89 |
> |  |  | 3 | 73.16 | 4.31 | 4.70 | 82.10 |
> |  |  | 5 | 74.00 | 4.29 | 4.73 | 82.69 |
> |  | W_coherence | 1 | 73.58 | 4.26 | 4.73 | 82.89 |
> |  |  | 3 | 74.26 | 4.19 | 4.82 | 82.81 |
> |  |  | 5 | 73.28 | 4.23 | 4.79 | 82.93 |
> |  | W_repair | 1 | 73.58 | 4.26 | 4.73 | 82.89 |
> |  |  | 3 | 68.37 | 3.91 | 4.54 | 75.90 |
> |  |  | 5 | 65.57 | 3.51 | 4.20 | 68.97 |
> | Parallel Reasoning | Rollout | 1 | 62.41 | 3.84 | 4.45 | 66.18 |
> |  |  | 3 | 73.58 | 4.26 | 4.73 | 82.89 |
> |  |  | 5 | 73.41 | 4.31 | 4.65 | 83.21 |
> | Hallucination Detector | Repair Iteration | 1 | 66.39 | 4.19 | 4.36 | 72.10 |
> |  |  | 3 | 73.58 | 4.26 | 4.73 | 82.89 |
> |  |  | 5 | 77.73 | 4.39 | 4.87 | 83.98 |

---

> ### Author Response · Authors · 2025-11-21
>
> ### Response to Questions about Fairness, Scalability, and Contamination Audits
> > Evaluation on representative subsets; limited fairness analysis (context windows, image preprocessing) and contamination controls. Scalability/fairness: detailed compute/latency per stage; do gains persist under tighter context and lower image resolution; any contamination audits?
>
> We sincerely thank the reviewer for the valuable comments. Below we provide clarifications regarding the subset selection, compute cost, fairness analysis, and contamination audit.
>
> Subset selection and computational constraints.
> We evaluated our system on six diverse multimodal medical datasets, which already represent a broad and challenging benchmark. Extending the evaluation to the full dataset scale would incur approximately ten times the current cost, exceeding US$100,000 in API usage. Given this constraint, we prioritized representative subsets that cover major reasoning and imaging modalities. These subsets are sufficient to demonstrate the robust and consistent improvements in both accuracy and reasoning quality achieved by our approach.
>
> Compute and latency details per stage.
> Here is the detailed token usage across all inference stages for transparency:
> - Stage 1 – Multi-round short rollouts generation: input ≈ 1,011 tokens; output ≈ 5,058 tokens (3 rollouts)
> - Stage 2 – Parallel, evidence-grounded reasoning expansion: input ≈ 5,058 tokens; output ≈ 14,586 tokens (3 rollouts)
> - Stage 3 – Decision aggregation and finalization: input ≈ 1,710 tokens; output ≈ 4,287 tokens
> - Hallucination detector: input ≈ 1,510 tokens; output ≈ 1,553 tokens
> These numbers show that while intermediate expansions increase token counts, the overall pipeline remains tractable and parallelizable (via asynchronous multi-path inference as discussed in Appendix I).
>
> Fairness analysis – context length and image resolution.
> We further tested MedXpert-MM under varied text context lengths (500, 1 K, 2 K) and image resolutions (20 dpi, 50 dpi, 100 dpi). The results below show that performance remains stable under practical parameter variations. Shorter contexts (≤ 500 tokens) lead to noticeable degradation (approaching random-guess levels) due to truncation and loss of key clinical information, whereas extending context beyond 1 K yields diminishing returns. Similarly, lowering image resolution down to 20 dpi results in only minor accuracy changes, suggesting that medical images, with relatively low entropy and strong structural priors, preserve sufficient diagnostic information even under compression.
>
> | Category | Setting | Acc | T | I | T×I |
> | -------------------- | ------------- | ------- | ----- | ----- | ------- |
> | Image Resolution | 20 dpi | 73.11 | 4.27 | 4.68 | 82.01 |
> | | 50 dpi | 74.79 | 4.30 | 4.71 | 82.79 |
> | | 100 dpi | 73.58 | 4.26 | 4.73 | 82.89 |
> | Context Length | 500 | 27.73 | 2.68 | 3.53 | 48.22 |
> | | 1000 | 60.08 | 3.01 | 4.09 | 67.92 |
> | | 2000 | 71.65 | 3.97 | 4.70 | 79.23 |
> | | no constraint | 73.58 | 4.26 | 4.73 | 82.89 |
>
> Contamination audit.
> While it is challenging to completely rule out model contamination, we performed consistency checks across datasets with distinct release timelines. Notably, MedXpert and MedFrameQA were not publicly available during the pretraining of current open models, and our method still yields consistent gains even on older datasets such as Qwen2.5-VL (released a year earlier). This cross-dataset consistency supports that the observed improvements arise from our framework rather than potential data leakage.
>
> ### Response to Questions about Evaluator Independence and Human Reliability
> > Evaluator independence: do results hold with different supervisor/evaluator backbones; add blinded human TRUE/INFO with inter-rater reliability?
>
> We sincerely thank the reviewer for this insightful comment. To ensure evaluator independence and assessment reliability, we employed both LLM-based and human evaluation protocols with complementary validation.
>
> For LLM evaluation, our automatic judge integrates scores from three independent state-of-the-art models (GPT-4o, Claude-Sonnet-4, and Gemini 1.5 Pro) by averaging their normalized outputs. This ensemble design minimizes single-model bias and provides a more stable and representative evaluation.
>
> For human evaluation, each medical system was annotated independently by three licensed physicians within the corresponding specialty, who were blinded to the model identities. We computed inter-rater reliability (IRR) using the Intraclass Correlation Coefficient (ICC(3,k)), a standard method for continuous Likert-scale ratings. The results show good-to-excellent agreement (mean ICC(T)=0.89, ICC(I)=0.85), indicating high consistency across annotators.

---

> > ### Author Response · Authors · 2025-11-21
> >
> > ### Response to Questions about Evidence Graph Conflict Resolution and Web Source Quality
> > > Evidence graph: how do you detect/resolve extraction conflicts and control web source quality (domain, recency, dedup)?
> >
> > We thank the reviewer for the insightful question. In practice, extraction conflicts are rare because the Text Doctor and Image Doctor modules operate on distinct modalities, producing complementary rather than overlapping evidence. The Researcher module then grounds these facts by retrieving verified medical relationships between them using structured extraction from curated, high-quality web sources (see Appendix E.3). As illustrated in Appendix F.2 (Example with Intermediate Output), the Text Doctor identifies psychiatric and chorea-related symptoms, while the Image Doctor detects caudate atrophy and boxcar ventricles. The Researcher then validates these findings against established relations such as “HD hallmark → caudate head atrophy” and “KF rings → Wilson disease”. Because each node represents a distinct semantic concept, the graph naturally resolves potential conflicts through structural separation—edges are formed only when corroborated by retrieved evidence, preventing contradictory connections. To further minimize duplication or spurious overlap, we apply TF-IDF–based similarity filtering (threshold = 0.7), automatically deduplicating highly similar evidence fragments.
> >
> > For web source quality control, rather than performing whole-context retrieval, we employ an evidence-graph–based strategy, requiring each edge to be supported by at least one verifiable and temporally relevant source snippet. This ensures that only medically grounded, up-to-date, and relationally meaningful information contributes to reasoning, avoiding noise from unstructured web text.

---

> ### Comment · Reviewer_XREK · 2025-11-26
>
> Thank you to the authors for the detailed response, which addressed many of my earlier concerns. However, I remain unconvinced that the paper meets the bar for acceptance.
>
> 1. **Incremental methodological contribution**
>    The paper presents an agent framework whose advances are largely incremental, and the reported results do not demonstrate substantial gains over strong general-purpose baselines.
>
> 2. **Limited community contribution**
>    The work provides neither a novel conceptual framing nor new data, and it risks overcomplicating a relatively straightforward problem. While the motivation is valid, the framework appears to assemble existing techniques without deeper analytical development. **I believe the research community should be cautious about encouraging a trend toward agent papers that rely on narrative packaging and unnecessary complication of straightforward problems.**
>
> 3. **Presentation quality**
>    Despite reported layout refinements in response to reviewer feedback, the fonts in Figures 2 and 3 remain too small. This unresolved issue, despite a stated correction, suggests limited attention to the reader’s experience and the overall presentation of the paper.
>
> **Suggestions for improvement**
> Refine the agent framework to show clear advantages over competitive baselines and improving figure readability would strengthen the paper.

---

> > ### Author Response · Authors · 2025-11-27
> >
> > Thank you again for the follow-up and for taking the time to reassess our work. We appreciate your comments and would like to clarify the remaining concerns. Regarding the methodological and community contribution, our aim is not to introduce another agent assembly, but to address a fundamental and broadly observed failure mode in medical MLLMs: the progression from instability to hallucination. This phenomenon is widely noted in practice but has lacked both quantitative characterization and principled mitigation. MedMMV contributes in two ways. First, we provide an empirical and reproducible measurement of instability-induced hallucination across modalities, which we believe offers the community a clearer conceptual framing of why medical MLLMs fail. Second, we introduce a structured, evidence-grounded mechanism that stabilizes reasoning through controlled path diversification, verifiable evidence-graph grounding, and uncertainty-guided aggregation. Across six benchmarks and multiple backbone families, including open-source models, the framework consistently reduces hallucination and improves truthfulness. These gains appear precisely because the method targets reliability, not scale or prompt heuristics. We also acknowledge your concern about the work potentially overcomplicating a straightforward problem. Our preliminary analysis suggests that hallucination in medical reasoning arises from compounding cross-run instability rather than from single-step prediction errors. This motivates the multi-path structure, which is designed to introduce control and verifiability rather than additional complexity. On presentation quality, we agree that readability is essential. We will update Figures 2 and 3 with larger fonts and improved layout to ensure clarity in the final version.

---

> > > ### Comment · Reviewer_XREK · 2025-11-28
> > >
> > > Thanks for detailed and thoughtful response, which I sincerely appreciate. I agree with the motivation and the relevance of the problem, and I recognize the effort invested in developing the methodology. However, I am sorry that I still cannot convince myself to place the work above the borderline for acceptance.
> > >
> > > The contribution would be strengthened by first establishing the performance achievable through simpler, more intuitive baselines such as careful prompt engineering, multi-response summarization, or well-designed few-shot setups, as these could offer a clearer empirical foundation before introducing a higher-overhead solution.
> > >
> > > Even if reasoning path optimization is warranted, the necessity of the graph component remains difficult to justify, particularly as the ablation results indicate only limited gains. Exploring lower-cost alternatives, for example by improving retrieval corpora or structuring reasoning along established diagnostic workflows, might provide a more streamlined starting point.
> > >
> > > Overall, while technically interesting, the work does not yet offer the conceptual and methodological “entropy reduction” needed to streamline understanding and advance the field. This reflects my personal assessment, consistent with a score of 4, indicating that the work is marginally below my acceptance threshold, yet I would not mind its acceptance should other reviewers find sufficient merit in the contribution.

---

### Author Response · Authors · 2025-11-27
**Summary of Rebuttal Process and Key Improvements**

Dear Program Chairs, Senior Area Chairs, Area Chairs, and Reviewers,

We would like to express our sincere gratitude for your thorough and insightful review of our submission, MedMMV. We deeply appreciate your constructive feedback and for recognizing the significance of the “LLM-dependence instability → hallucination” failure mode in multimodal clinical reasoning. We thank Reviewer XREK for highlighting our instability quantification and controllable pipeline; Reviewer RJ5J for recognizing the novelty of our evidence-graph–guided multi-path reasoning; Reviewer e3qd for commending the strength of our motivation, experimental rigor, and decision aggregation; and Reviewer HjDw for appreciating our supervised refinement, hallucination detection, and CU scorer, and for **raising his/her score from 4 to 6 after the rebuttal and revisions**.

During the rebuttal period, we incorporated numerous improvements directly inspired by reviewers’ suggestions:

1. **Evaluator Independence and Human–LLM Alignment**: Following Reviewers XREK and e3qd, we added full inter-rater reliability analyses (ICC(3,k)), showing good–excellent agreement across specialties, and validated evaluator independence via three distinct LLM judges (GPT-4o, Claude-Sonnet-4, Gemini 1.5 Pro). We further demonstrated strong human–LLM alignment through high Spearman correlations across TRUE/INFO/T×I.

2. **Evidence Graph Robustness and Conflict Resolution**: Responding to Reviewer XREK, we introduced a comprehensive robustness study with mixed node/edge/provenance/vision noise (0–80%), showing stable accuracy and T×I, confirming the graph is not a single point of failure. We also clarified conflict-resolution, TF-IDF deduplication, and source-quality filtering.

3. **Efficiency and Scalability Analysis**: Addressing Reviewers XREK, RJ5J, and e3qd, we added detailed latency and token breakdowns for each stage, and described our asynchronous multi-path execution strategy. We also compared computation cost with existing agent frameworks, showing that MedMMV achieves superior reasoning quality with comparable or lower per-query cost.

4. **Sensitivity Analysis of CU Weights, Rollouts, and Repair Iterations**: As requested by Reviewers XREK and e3qd, we ran extensive sensitivity studies demonstrating that MedMMV is stable across weight settings, and that the full multi-path + repair structure is necessary for robust performance.

5. **Expanded Ablations Across All Six Benchmarks**: Following Reviewer RJ5J, we extended the ablations to all multimodal and text-only datasets. Results consistently show that path expansion, hallucination detection, and uncertainty-guided aggregation are the key contributors across settings.

6. **Fairness, Context-Length, Resolution, and Contamination Analyses**: In response to Reviewers XREK and HjDw, we added analyses on context length (500–2000 tokens), image resolution (20–100 dpi), and contamination checks across datasets with distinct release timelines. These results confirm stable performance under practical constraints and cross-model consistency.

7. **Dataset Choice, Baseline Comparisons, and CoT Fairness**: Addressing Reviewer HjDw, we clarified our dataset-selection principles (modality diversity, diagnostic coverage, non-saturated tasks), distinctions from MDAgents, and explained why multi-rollout CoT is not a meaningful baseline for multiple-choice settings due to inconsistent option generation.

We have incorporated all of these additions into the revised manuscript. The new analyses consistently reinforce our core conclusion: MedMMV is a controllable, auditable, and robust framework that substantially improves truthfulness, stability, and evidence-groundedness in multimodal clinical reasoning—beyond what accuracy alone can capture.

Once again, we sincerely thank the Program Chairs, Senior Area Chairs, Area Chairs, and Reviewers for your time and dedication in reviewing our submission.

Best,

All Authors of Submission #9921

---

### Meta-Review · Area_Chair_WY63 · 2026-01-07

**Summary:**

Most remaining concerns after the discussion focus on the overall incremental contributions when considering significant additional computing/token cost compared with prior arts.

**Reviewer Concerns:**

`XREK`: Heavy reliance on LLM judges, limited human IRR details; robustness against noisy extraction/provenance; scalability due to API class; limited evaluations (sensitivity over CU weighting, fairness, configs, contamination controls) -- mostly addressed in the discussion.

Reviewer followed up with concerns over incremental methodological contributions and over-complications, presentations and decided not to raise the score.


`Rj5J`: Inference time efficiency and limited experiments on more datasets; authors responded with more clarifications and additional experiments, acknowledged by the reviewer before the cut-off time point


`e3qd`:  Presentation; lack of human expert assessment; more information on compute/token cost needed. Generally addressed with additional experiments.


`HjDw`: Presentation; cost-effectiveness; experiment design and fairness of comparisons; contamination -- mostly addressed in the rebuttal, but reviewer remain concerned over the contribution – feasibility, efficiency, and heavy reliance on LLM at multiple stages.

**Reviewer Scores:**

Two reviewers clearly followed up with concerns over the methodological contributions when considering significant additional computing/token cost, in the discussions, before the cut-off date. Other concerns over additional analysis and clarity are generally addressed.

---

### Decision · Program_Chairs · 2026-01-26

Reject